



# biogeodyn-MITgcmIS (v1): a biogeodynamical tool for exploratory climate modelling

Laure Moinat[1,2,3], Florian Franziskakis[2], Christian Vérard[3,4], Daniel N. Goldberg[5], and
Maura Brunetti[1,2,3]

[1]Group of Applied Physics, University of Geneva, Rue de l'École de Médecine 20, 1205 Geneva, Switzerland
[2]Institute for Environmental Sciences, University of Geneva, Bd. Carl-Vogt 66, 1205 Geneva, Switzerland
[3]Centre pour la Vie dans l'Univers (CVU), University of Geneva, Geneva, Switzerland
[4]Section of Earth and Environmental Sciences, University of Geneva, Rue des Maraîchers 13, 1205 Geneva, Switzerland
[5]School of GeoSciences, University of Edinburgh, Edinburgh, UK

**Correspondence:** Laure Moinat (laure.moinat@unige.ch)

**Abstract.** Modelling the climate system is challenging when slow-response components, such as the deep ocean, vegetation and ice sheets, must be evolved alongside fast-response ones. This is crucial for investigating, for example, climate tipping elements and their interactions on the global spatial scale over multimillennia timescales. While Earth system models, such as those used in the Coupled Model Intercomparison Project (CMIP), are too computational expensive for simulations spanning thousands of years, simplified parameterizations and coarse resolutions in Earth Models of Intermediate Complexity (EMICs) can significantly affect the nonlinear interactions among climate components. Here, we describe a new tool, *biogeodyn-MITgcmIS*, which has a complexity level intermediate between EMICs and CMIP-class models. The core of *biogeodyn-MITgcmIS* is a coupled MITgcm setup that includes atmosphere, ocean, thermodynamic sea ice, and land modules. To this, we have added offline couplings with a vegetation model (BIOME4), a hydrological model (pysheds), and a new global-scale ice sheet model (*MITgcmIS*). The latter is implemented on the same cubed-sphere grid as MITgcm, using the shallow-ice approximation, as well as MITgcm outputs and a modified Positive Degree Day method to estimate the ice-sheet surface mass balance. Here, we describe in detail the new ice sheet model and the coupling procedure. We evaluate *biogeodyn-MITgcmIS* using simulations for the pre-industrial period and the 1979-2009 period. These two experiments allow us to assess the model's performance against CMIP-class models, as well as a combination of reanalyses and observations. *biogeodyn-MITgcmIS* successfully reproduces the large-scale climate and its major components, with results comparable to those of two CMIP models with dynamical vegetation. We discuss its potential applications and future developments.

## 1 Introduction

As the atmospheric $CO_2$ concentration raises under the present-day climate crisis, there is an increasing risk of crossing critical thresholds where some parts of the climate system suddenly change, potentially in an irreversible manner. Many elements of the Earth climate (denoted as 'tipping elements) can indeed experience abrupt changes in their dynamics (Lenton et al., 2008; McKay et al., 2022). Some examples were reported from proxy data related to past climate regime shifts, such as the Atlantic



Meridional Overturning Circulation (AMOC) or ice sheets during the glacial cycles (Boers et al., 2018; Brovkin et al., 2021). Tipping behavior in the climate system is very difficult to study because changes occur over a large range of temporal scales (from decades in the Amazon forest to centuries in AMOC or multimillennia for the complete melting of ice sheets), at local or
global spatial scales (Wunderling et al., 2024). Causal relationships giving rise to tipping cannot be easily inferred from sparse proxy data, and numerical simulations are often not able to span all the timescales because of too high computational costs.

There are several numerical techniques that have been proposed for studying tipping dynamics, from regional to global circulation models, at different levels of complexity in process representation, depending on the selected temporal and spatial scales (Wunderling et al., 2024). We are interested in a modeling setup that can span daily to millennial timescales at the global
spatial scale, thus including at least atmosphere, surface and deep ocean, sea ice, land vegetation and ice sheet dynamics. While these components are sometimes included in CMIP-like models (Eyring et al., 2016), these models require high computational costs to run for thousands of years and thus simulations cannot reach stationarity in slow deep-ocean and ice sheet dynamics (Balaji et al., 2017). In contrast, Earth Models of Intermediate Complexity (EMICs) are fast reaching a stationary state (Claussen et al., 2002; Willeit et al., 2022), but their coarse spatial resolutions and simplified parameterizations can have
non negligible impacts on the nonlinear interaction among climatic components.

Reaching stationarity is an important property for characterizing climatic attractors, i.e. control runs describing the asymptotic state of all climatic variables under a given forcing and unveiling the nonlinear backbone structure of the climate dynamics (Hawkins et al., 2011; Drótos et al., 2017; Lucarini and Bódai, 2017; Brunetti et al., 2019; Brunetti and Ragon, 2023). Once the attractors are found, transient simulations can also be run using different kinds of perturbation (random noise or
deterministic scenarios) to investigate shifts from one stationary state to another (Lucarini and Bódai, 2020; Lucarini et al., 2022; Moinat et al., 2024). In particular, these transient simulations under decreasing and increasing forcing can reveal not only tipping dynamics over millennial time scales, but also domino effects and process reversibility at local or global spatial scales. However, it is crucial that all climatic components evolve in a consistent way under the same forcing, including those with slow response, such as deep ocean, ice sheets and terrestrial vegetation, which have been identified as potential tipping
elements (Wunderling et al., 2024).

Nowadays, there are few CMIP6 models with interactive ice sheets (Ackermann et al., 2020; Muntjewerf et al., 2020; Madsen et al., 2022) and/or dynamical vegetation (Drüke et al., 2021). However, as said above, high computational costs of this type of models makes them only adapted for studying climate evolution on centennial timescales. A technique for speeding up complex models and thus enabling them to explore feedbacks on longer timescales is offline (or asynchronous)
coupling (Claussen, 1994; Herrington and Poulsen, 2011; Pohl et al., 2016). This is particularly useful for including tipping elements with a slow response. In practise, the climate is first estimated with fixed ice sheets and vegetation, and eventually the latter are updated to match the equilibrium conditions of the former (Foley et al., 1998). The procedure can be repeated until convergence. In this way, fast and slow components evolve consistently until a stationary state is reached. Similar steps can be used in transient simulations between one climatic steady state and another by increasing the coupling frequency.
Here, we propose a simulation tool, called *biogeodyn-MITgcmIS*, that combines surface processes, biology adaptation, and climate dynamics on multimillennial timescales with a complexity that is in between EMICs and CMIP-like models. Such





coupled setup allows one to take into account in a consistent manner the evolution of major processes of the Earth over a multi-millennial time scale: vegetation, atmosphere, ocean, cryosphere, hydrosphere and their interactions, under different boundary conditions given by continental and oceanic configurations. We will provide a complete description of all the components and

in particular the newly developed ice sheet model (*MITgcmIS*). To assess the capability of *biogeodyn-MITgcmIS* to represent the present climate, we will validate its results against those obtained using some CMIP6 class models and reanalysis products.

## 2    Biogeodynamical tool

Here, we describe each component of our biogeodynamical tool, including the boundary conditions and the coupling strategy.

### 2.1    GCM - Coupled MITgcm setup

The dynamical core of our tool is given by the MIT general circulation model (MITgcm) (Marshall et al., 1997a, b; Adcroft et al., 2004), which solves the Navier-Stokes equations for the atmosphere and the ocean over the same cubed-sphere (CS) grid (Marshall et al., 2004). In particular, we consider MITgcm (version c68s) in a coupled setup including atmosphere, ocean, thermodynamic sea ice and land. In our simulations, we use the so-called CS32 configuration, where each face of the cube has $32 \times 32$ grid cells, corresponding to an average horizontal resolution of 2.8°. This or similar setups have been already

used for studying idealised configurations such as the coupled aquaplanet (Ferreira et al., 2011; Rose, 2015; Ferreira et al., 2018; Brunetti et al., 2019; Ragon et al., 2022; Zhu and Rose, 2023; Brunetti and Ragon, 2023; Moinat et al., 2024), deep-time climates (Brunetti et al., 2015; Ragon et al., 2024, 2025) and the present-day climate (Brunetti and Vérard, 2018).

Physical parameterizations for the atmosphere are based on the 5-layer SPEEDY model (Simplified Parameterizations, privitivE-Equation DYnamics) (Molteni, 2003). SPEEDY is in the intermediate complexity class, and includes simplified rep-

resentations of convection, large-scale condensation, vertical diffusion, surface fluxes of momentum and energy. The radiative scheme uses two spectral bands for the shortwave radiation and four for the longwave radiation. Cloud cover and thickness are defined diagnostically from the values of relative and absolute humidity. Cloud albedo depends on latitude, as done in Ragon et al. (2022), for reducing the net solar radiation at high latitudes and therefore having a better agreement with observational data (Kucharski et al., 2013). Five pressure levels are represented, going from 1000 hPa to 0 Pa, where the bottom level rep-

resents the planetary boundary layer, the top one the stratosphere, and the remaining three the free troposphere. SPEEDY has been evaluated against NCEP-NCAR and ERA5 reanalysis (Molteni, 2003) and, despite its simplified parameterizations, has been assessed to provide a realistic description of the atmosphere, with the advantage of requiring less computer resources than state-of-the-art Atmospheric GCMs. A simple 2-layer land model (Hansen et al., 1983) is coupled to SPEEDY.

The physics packages activated for the oceanic component are the KPP scheme (Large and Yeager, 2004) to account for ver-

tical mixing in the water column, and the Gent and McWilliams scheme  (Gent and McWilliams, 1990) to describe mesoscale eddies. The Winton model (Winton, 2000) is used to include sea ice thermodynamics (THSICE), while sea ice dynamics is neglected. In our setup, there are 25 vertical nonuniform levels in the ocean, with thickness ranging between 20 m near the surface and 1300 m at the bottom.



The coupled MITgcm setup needs to have the following input files: bare-surface albedos, vegetation fraction, bathymetry,
topography, runoff map, salinity and sea temperature for all the ocean levels. Orbital parameters can be set by specifying the
obliquity, the duration of the day and the radiative influx from the sun. It runs about 200 years per CPU day using 25 cores.

## 2.2 Boundary conditions - land-ocean configurations

Our tool can be applied for describing the present-day as well as deep-time climates. While the continental configuration for the
present day is well known and based on observations, we need to use paleogeography reconstructions for deep-time climates.
Several options are available (Scotese, 2021; Merdith et al., 2021; Vérard, 2019). Even if in previous works the MITgcm setup
was coupled to the PANALESIS reconstructions  (Brunetti et al., 2015; Brunetti and Vérard, 2018; Ragon et al., 2024, 2025),
it is important to note that our tool can be started from any boundary conditions (i. e., alternative paleogeographical recon-
structions, idealised land-ocean configurations or aquaplanet) and ocean depths, which is useful for exoplanet or conceptual
studies.

In both cases of present-day and deep-time climates, the procedure for adapting the high resolution geographical maps to the
MITgcm grid will be the same. Either using the present-day ETOPO global relief model (ETOPO, 2022) or a paleogeographic
reconstruction from PANALESIS, the input geography is given in the latitude/longitude coordinates system with arc-sec hori-
zontal resolution. Since the simulations are performed with the cubed-sphere CS32 grid, a smoothing procedure is needed to
upscale to the 2.8° resolution. Then, we need to remove isolated oceanic points or lakes independent of their size, since the
MITgcm becomes numerically unstable when there are enclosed areas of water. Moreover, very shallow waters are also prone
to instability, especially when sea ice develops; therefore, we set all oceanic points shallower than $-20$ m to this depth.

## 2.3 Vegetation - BIOME4

BIOME4 is a vegetation model that predicts the global steady state of the vegetation distribution corresponding to long-term
averages of monthly mean surface air temperature (SAT), sunshine and precipitation (Kaplan, 2001). Additional inputs are soil
depth and texture, which are used to determine water holding capacity and percolation rates (Kaplan et al., 2003). Moreover,
the atmospheric $CO_2$ content needs to be specified. All these quantities are obtained from steady-state simulations performed
using MITgcm in the coupled configuration described in Sec. 2.1.

BIOME4 follows the principle that ecosystems can be divided into a set of biomes characterised by the performance of plant
functional types (PFTs) (Haxeltine and Prentice, 1996; Kaplan, 2001). The model selects among a set of 12 PFTs the subset
that can be present in a grid cell on the basis of physiological and climatic constraints, like minimal temperature and water
supply. Using a coupled carbon and water flux model, BIOME4 calculates the net primary productivity (NPP) of each PTF
and the corresponding seasonal maximum leaf area index (LAI) that maximises NPP. At this point, competition among PTFs
is simulated by selecting the PTF with the optimal NPP as the dominant plant type. Opposing effects due to light competition
and wildfires are included through semi-empirical rules. The final output is the vegetation distribution in terms of the dominant
and secondary PFTs, total LAI and NPP, which can be classified into biome types, for a total of 27 biomes (28 including land
ice).




Main differences between BIOME4 and earlier versions (developed by Prentice et al. (1996); Haxeltine and Prentice (1996)) are the inclusion of new PTFs to represent vegetation types in the polar regions, and the calculation of photosynthetic pathways (for $C_3$ and $C_4$ plants) that depend on the PTF. BIOME4 and its earlier versions have been used to investigate climate-biosphere
interactions in the past (de Noblet-Ducoudré et al., 2000; M. Haywood et al., 2002; Kaplan et al., 2003; Salzmann et al., 2008; Sellwood and Valdes, 2008; Ragon et al., 2024, 2025). BIOME4 has also been used to assess the impact of current climate changes on the distribution of vegetation types (Allen et al., 2024).

## 2.4  Ice sheet - *MITgcmIS*

We have developed a Python code, called *MITgcmIS*, that describes the evolution of ice sheets at the global scale on the same
cubed-sphere grid used by MITgcm. Since we are interested in horizontal resolutions of the order of $2°$ or coarser, we can neglect basal melting and other fine-scale processes, as calving and ice streams. Note that there is already an MITgcm module, called STREAMICE, which implements in Fortran these small-scale processes at the km-scale (Goldberg and Heimbach, 2013).

In *MITgcmIS*, we use the shallow-ice approximation (Cuffey and Paterson, 2010) to model the ice-sheet movement and an
improved Positive Degree Day (PDD) (Braithwaite, 1977) method to compute the Surface Mass Balance (SMB), as we also want to apply our coupled framework to paleoclimates.

We choose an improved PDD approach instead of the Surface Energy Balance (SEB) method, since the latter requires quantities at the km-scale to estimate the energy budget, such as layer structure, surface roughness, and stability of the surface terrain to obtain latent and sensible heat fluxes (Hock and Holmgren, 2005). Thus, the SEB method is generally used in regional
climate models, which are able to reach the required accuracy in the representation of the climatic fields (especially clouds), in general provided by reanalyses (Wake and Marshall, 2015). This is an important limiting factor to consider that makes it difficult to apply SEB to paleoclimate simulations.

Although the PDD approach succeeds in representing the SMB in coarse simulations, it can underestimate the melting in past periods of high insolation (Plach et al., 2018). Moreover, since this method does not account for energy exchanges between the
ice sheet and the other components of the climate system, it cannot ensure a closed energy budget. Despite its limitations, we believe that the PDD method is the best choice when the representation of the atmosphere is at the global scale and simplified as in the SPEEDY module (Sec. 2.1).

### 2.4.1  Basic description

We start from the equation that describes the depth-integrated mass conservation for incompressible ice (Schoof and Hewitt,
150  2013):

$$\frac{dH}{dt} = -\boldsymbol{\nabla} \cdot \boldsymbol{q} + \dot{A} - \dot{B} \tag{1}$$

where $H = z_S - z_B$ is the height of the ice sheet between the bed $z_B$ and the surface $z_S$ vertical coordinates, $\boldsymbol{q} = \int_{z_B}^{z_S} \boldsymbol{U} dz$ is the horizontal flux obtained by vertical integration over the ice thickness of the horizontal part $\boldsymbol{U}$ of the velocity vector,





$\dot{A}$ is the SMB rate and $\dot{B}$ is the basal melting rate. In our case, we neglect the basal melting rate, hence $\dot{B} = 0$. Eq. (1) is
simply an expression of mass continuity; but the specific form of $\boldsymbol{q}$ derives from the full Stokes equations in the so-called
shallow-ice approximation, based on the low aspect ratio ($\sim 10^{-2}$-$10^{-3}$) between vertical and horizontal length in ice sheets.
This relationship describes how ice flux responds to ice-sheet geometry, as described below.

Ice rheology can be simplified by neglecting the ice bed movement, by considering that the only important type of deforma-
tion is vertical shearing, and by assuming a power-law shear thinning viscous rheology, with strain rates $\dot{\epsilon}$ proportional to the
driving stress $\tau_d$, as $\dot{\epsilon} = a\tau_d^n$, where $a$ is the Glen's law coefficient, which is assumed constant, and $n$ is typically set to 3. This
leads to the following relation (Cuffey and Paterson, 2010):

$$q = \frac{2a}{n+2}\tau_d^n H^2 = -D\nabla z_S \tag{2}$$

where $\tau_d = \rho_i g H \alpha$, $\alpha = -\nabla z_S$ is the surface slope, $g$ the gravitational acceleration, $\rho_i = 920\,\mathrm{kg\,m^{-3}}$ the ice density, and

$$D = \frac{2a}{n+2}(\rho_i g)^n |\nabla z_S|^{n-1} H^{n+2} \tag{3}$$

By combining the two relations, one obtains

$$\frac{dH}{dt} = \nabla \cdot (D\nabla H) + \nabla \cdot (D\nabla z_B) + \dot{A} \tag{4}$$

Eq. (4) is numerically integrated in Python on the cubed-sphere grid, once the SMB term $\dot{A}$ is determined, as detailed in the
next sections.

As noted above, we do not consider spatially varying Glen's law parameter $a$ or basal sliding. Both processes are sometimes
included in modelling of paleo ice sheets, by employing thermomechanical components to model ice temperature (which
influences Glen's law, Cuffey and Paterson (2010)), and to determine where basal sliding occurs due to thawed-bed conditions
(e.g., Moreno-Parada et al., 2023). However, the coarse resolution of our numerical grid does not allow representing of the
fast streaming that results from basal melting. Since our purpose is to investigate climatic steady states in which ice sheets
are in balance with the ocean, atmosphere, and biosphere, we choose not to represent these higher-order processes. Moreover,
the inclusion of such processes would introduce additional uncertain quantities and formulations – such as the temperature
dependence of $a$, the pattern and magnitude of geothermal heat flux, and the response of basal stress to basal water formation
and drainage. Instead, using a single parameter, $a$, to describe ice-sheet dynamics allows it to be constrained straightforwardly
using ice volume, as shown in the results section.

### 2.4.2 Surface mass balance - ablation

To compute the SMB we use a method based on the Positive Degree Day (PDD), which has in addition a percolation layer for
correctly assessing the melting (Tsai and Ruan, 2018).

The idea of the improved PDD method is to account for the presence of a percolation layer of thickness $H_p$ that creates a
delay in melting at the ice surface, as the ice is not expected to melt as soon as the SAT reaches $0°C$ (Tsai and Ruan, 2018). The
heat is assumed to be diffused downwards in the percolation layer, which reaches a uniform temperature $T_p$ on a relatively fast



timescale. In our setup, we assume that the percolation depth $H_p$ is constant even if in reality it can slightly change, depending on the type of ice. Below the percolation depth, the temperature quickly relaxes to an equilibrium value that does not depend on diffusion (Tsai and Ruan, 2018).

We assume that, between the ice surface ($z = 0$) and the height where the temperature measurement is performed ($z = h$), there is a constant temperature gradient. Thus, the heat flux can be written as:

$$q(t) = -k\frac{\partial T}{\partial z} = -k\frac{T_a(t) - T_p(t)}{h} \tag{5}$$

where $k$ is the effective thermal conductivity of air, $T_a$ is the temperature at $z = h$, and $T_p$ is the percolation layer temperature. From this assumption, and using conservation of energy between the percolation layer and the air, one can compute the ODE for the percolation layer temperature and for the ablation rate $a_r$ (Tsai and Ruan, 2018):

$$\frac{dT_p}{dt} = \frac{k}{h\rho_i c_P H_p}(T_a - T_p) \qquad \text{if } T_p < 0 \text{ or } T_a < 0 \tag{6}$$

$$a_r(t) = -\frac{dz_s}{dt} = \frac{k}{h\rho_i L}(T_a - T_p) \qquad \text{if } T_p = 0 \text{ or } T_a > 0 \tag{7}$$

where $z_s$ is the ice surface elevation. Constants $k$, $h$, $\rho_i$, $c_P$, $H_p$ and $L$ are all assumed to be known. We have used the same values as in Tsai and Ruan (2018), which result from a present-day calibration, namely $\rho_i = 920 \text{ kg m}^{-3}$, $c_P = 2100 \text{ J kg}^{-1}\,{}^\circ\text{C}^{-1}$, $k/h = 24 \text{ W m}^{-2}\,{}^\circ\text{C}^{-1}$, $L = 334 \text{ kJ kg}^{-1}$, $H_p = 10 \text{ m}$, $h = 2 \text{ m}$. The required input is the air surface temperature $T_a$. Tsai and Ruan (2018) showed good agreement with observations, along with a significant improvement in capturing early-season melting compared to the classical PDD method. During each model iteration, we compute the SMB by including a lapse rate correction that accounts for changes in surface elevation (see Sec. 2.4.4).

### 2.4.3 Surface mass balance - accumulation

To determine the remaining contribution to the SMB, we evaluate the accumulation of snow. This quantity is obtained by using outputs from the MITgcm. To be consistent with the energy budget in the MITgcm, the accumulation is estimated from the snow precipitation per grid cell. Since the MITgcm land module does not include accurate snow physics and, in particular, a process that densifies snow over time to obtain glacial ice, this densification is assumed to happen instantaneously. Therefore, as snow precipitation is expressed in units of [kg m$^{-2}$ s$^{-1}$], we divide this quantity directly by the glacial ice density, $\rho_i = 920 \text{ kg m}^{-3}$ to obtain the accumulation rate in [m s$^{-1}$].

In summary, two MITgcm outputs are needed: the surface air temperature (for the ablation) and the snow precipitation (for the accumulation). These two quantities are extracted from a simulation that has reached a steady state. We take daily outputs over an interval of 30 years, and then we take the average of these quantities for each day and each grid cell. Finally, the SMB is obtained by subtracting the ablation from the accumulation rates, and inserted at the right-hand side of Eq. (4), which is then solved in terms of the ice thickness $H$.





### 2.4.4 Isostatic, lapse-rate, freshwater and sea-level corrections

Taking into account the isostatic correction due to the ice sheet mass can be computed in several ways. Here, we adopt the Local Lithosphere Relaxing Asthenosphere (LLRA) method, where a time delay is included (Greve and Blatter, 2009). The idea behind this method is that there is a displacement of the lithosphere in the $z$ direction by $w_{ss}$, which is due to the ice load. A steady state is reached when the buoyancy force equilibrates the ice load (Greve and Blatter, 2009):

$$\rho_a g w_{ss} = \rho_i g \Delta H_{\text{ice}} \tag{8}$$

where $\rho_i = 920\,\text{kg}\,\text{m}^{-3}$ is the ice density, $\rho_a = 3300\,\text{kg}\,\text{m}^{-3}$ is the density of the asthenosphere and $\Delta H_{\text{ice}}$ is the ice thickness calculated by the ice sheet model. Thus

$$w_{ss} = \frac{\rho_i}{\rho_a} \Delta H_{\text{ice}} \tag{9}$$

However, the response of the asthenosphere is not immediate due to its viscous properties, and has a time delay that can be parameterised as:

$$\frac{dw}{dt} = -\frac{1}{\tau_a}(w - w_{ss}) \tag{10}$$

where $\tau_a$ is typically set to 3000 years (Greve and Blatter, 2009). At the end of each iteration $\Delta t$ of *MITgcmIS*, the ice sheet elevation is computed as:

$$H_{\text{total}} = H_{\text{topo}} + \Delta H_{\text{ice}} - \frac{dw}{dt}\Delta t \tag{11}$$

where $H_{\text{topo}}$ is given by the topography file.

As the surface elevation is varied when an ice sheet develops by $\Delta z$, not only the topography changes but also the SAT. Thus, a correction needs to be included by estimating the lapse rate $dT/dz$, as follows:

$$T_{new} = T_{pickup} - \frac{dT}{dz}\Delta z \tag{12}$$

where the lapse rate is computed at each ice-sheet grid point using the MITgcm output of the first run.

  Finally, since some new ice sheets formed or disappeared, the amount of water that has been exchanged with the ocean is
estimated. For each iteration the freshwater flux to or from the ocean is computed and included at restart at the ocean boundary of the ice sheet. To guarantee the conservation of salt, a compensation is performed at the global scale, as it is done in AMOC hosing experiments (Jackson et al., 2023). The variation of water volume in the ocean is converted in sea level change, with updated coastlines defining new topography (including ice sheet height), mask and bathymetry files.

### 2.5 Runoff - pysheds

In our study, we need to consider different continental configurations corresponding to the Earth's evolution, under a range of ice sheet loading. Hence, for each new configuration, we need to recalculate the runoff map. The MITgcm needs as an input a





file with three arrays, specifying for each land point $L_i$ the corresponding precipitation storage area $A_i$ and the ocean point $O_i$ where it is drained (outlet point).

For present-day as well as for past (palæo-) topographies, we discriminate between land and ocean points by defining a
contour line at 0 m in elevation. If any area with negative values are fully enclosed within area with positive values, they are considered as 'lakes' unless elevation reaches values below $-2000$ m. In such cases, we re-assign the elevation in order to remove the depression and reroute the flow direction. For this purpose, as well as for cleaning local pits, depressions and flat terrains are corrected using the fill_pits, fill_depressions, and resolve_flats functions from pysheds (Bartos, 2020). This ensures a continuous Digital Elevation Model (DEM) with no single pixel or stagnant areas where water would not flow. Finally, we
clip the DEM to all elevations above 0 m and retrieve the closest outlet point.

Every point in the MITgcm grid is hence defined as 'continental' or 'oceanic' depending on whether or not it is located inside the land (positive elevation) or not. Using the corrected topography, we generate a flow direction by applying an eight-direction (D8) flow routing algorithm from pysheds. This method assumes that water from each cell in the DEM will flow to one of its eight neighboring cells, the one that results in the steepest descent. The D8 algorithm is computationally efficient and widely
used for hydrological modelling.

The slope from a cell $c$ to each of its eight neighbors $i$ is calculated as:

$$s_i = \frac{Z_c - Z_i}{d_i} \tag{13}$$

where $Z_c$ is the elevation at the center cell, $Z_i$ is the elevation of the $i^{\text{th}}$ neighbor, and $d_i$ is the distance to that neighbor. For cardinal directions (N, E, S, W), $d_i = 1$, and for diagonal directions (NE, SE, SW, NW), $d_i = \sqrt{2}$.

Then, the flow direction is determined by selecting the neighbor with the maximum positive slope:

$$\text{flow\_dir} = \arg\max_i (s_i), \quad \text{where } s_i > 0 \tag{14}$$

The resulting direction is encoded using a directional mapping:

$$[N, NE, E, SE, S, SW, W, NW] = [64, 128, 1, 2, 4, 8, 16, 32]$$

Each cell is assigned one of these values in the resulting flow direction map, indicating the direction water would flow from that cell based on the steepest slope. We then trace the flow path for every continental point using the flow direction until we reach the ocean, and define the nearest oceanic point as its outlet. Each initial continental point ultimately is being assigned
one oceanic outlet, while initial oceanic points are their own outlet. Note that this approach mimicking the drainage system is a great improvement regarding the water cycle relative to many climate models where the outlet of a drop of water is merely assigned to the closest ocean point.

Moreover, since the MITgcm has no proper online ice sheet model, excess water that would accumulate to form ice sheets is instead evacuated via runoff. More precisely, in the MITgcm code, snow precipitation $P_{\text{snow}}$ exceeding the tolerated limit
(usually set to 10 m) is automatically redirected into the ocean via the runoff. This creates an artificial excess of runoff in our asynchronous coupling, where $P_{\text{snow}}$ is now used in the SMB accumulation term, and hence a correction is necessary. Thus,





we introduce the following correction in the precipitation storage area $A_i$ at each land point:

$$A'_i = A_i \left( 1 - \frac{P^i_{\text{snow}}}{P^i_{\text{tot}} + P^i_{\text{snow}}} \right) \tag{15}$$

where $P^i_{\text{tot}}$ is the total rain precipitation at the land point $i$ and $P^i_{\text{snow}} \leq P^i_{\text{tot}}$. This correction has an effect only on land points
where the ice sheet is developing.

## 2.6    Coupling framework

Offline coupling between the coupled MITgcm atmosphere-ocean-sea ice-land setup and BIOME4 has been already success-
fully applied in Ragon et al. (2024, 2025). Here, we will document the comprehensive framework that includes BIOME4,
the new ice sheet module *MITgcmIS* and the runoff map calculation for different boundary conditions (present-day, paleo or
idealised configurations), as schematically illustrated in Fig. 1.

A first simulation is run until the coupled MITgcm setup has reached a steady state, defined by having a surface energy
balance $F_s < 0.2 \, \text{W/m}^2$ (usually several thousands of simulated years are required). Afterwards, two simulations with different
output frequencies are run, one with monthly frequency (for BIOME4) and the other with daily frequency (for *MITgcmIS*).

At this point, the offline coupling workflow can start. Before running the ice sheet model, the following correction are
required. For representing an advancing ice flow over shallow ocean, we mimic this process as follows: if the sea ice thickness
is equal to the ocean depth, the ocean point becomes a land point and hence the ice sheet can develop on it (up to $-20$ m of
depth). Then, the corrected topography file is given as an input to the ice-sheet model, together with daily MITgcm outputs
per grid cell for SAT and snow precipitation, which are used for calculating ablation and accumulation rates, respectively.
*MITgcmIS* is run for an equivalent of 40-100 thousand years, until the ice sheet reaches a steady state (MacAyeal, 1997). The
isostatic, lapse-rate and sea level corrections (Sec. 2.4.4) are included at this stage, giving rise to new topography (including
ice sheet height), mask and bathymetry files. Afterwards, pysheds is applied using these files in input, closing lakes and small
passages if necessary, and giving in output a new runoff map.

Finally, the vegetation model is run based on the new files. The outputs needed from MITgcm and the ice-sheet model
(namely, precipitation, SAT (corrected by the lapse rate) and sunshine) are converted on a latitude/longitude grid and then
given to the BIOME4 model. The equilibrium biome distribution is converted in new files for vegetation fraction and surface
albedo using the values reported in Haywood et al. (2010). Due to the coordinate change, some land and ocean points can be
inverted. Hence, a vegetation fraction equal to 0 and an albedo value equal to the default water value of 0.07 are assigned to
new ocean points on the CS grid, while the value of the closest land point is assigned to new land points.

Before running a second iteration, the pickup files that are used to restart a MITgcm simulation need to be updated to the
new values of SAT (including the lapse-rate correction). Then, the new pickup files, along with the new vegetation fraction,
albedo, topography, bathymetry and mask files, are given back to the MITgcm to run the whole coupling process at least twice,
so that the GCM has time to adjust to the new input files. The whole procedure forms the coupled *biogeodyn-MITgcmIS* model,
which thus describes in a consistent way the evolution of all the climatic variables, including those with a slow response, like
deep-ocean dynamics, vegetation and ice sheets.





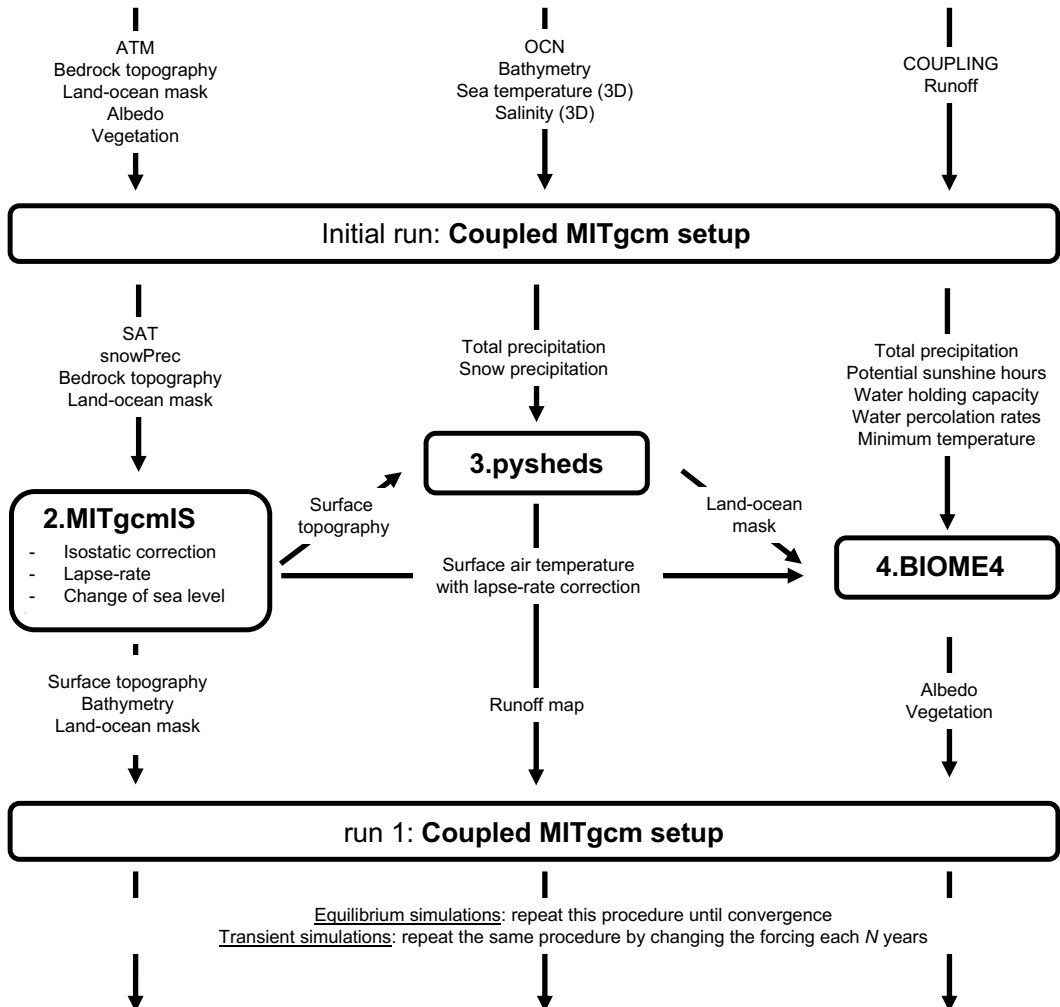

**Figure 1.** Schematic representation of the offline coupling framework.

## 3 Results

### 3.1 Validation procedure

#### 3.1.1 *biogeodyn-MITgcmIS* initial conditions

To start the first run of our simulations, it is necessary to provide initial conditions that are representative of the present-day climate. The three-dimensional distributions of sea potential temperature and salinity are derived from the Levitus World Ocean Atlas (Levitus, 1982). Orbital forcing is prescribed at present-day values, with a solar constant of 1368 W m$^{-2}$ and obliquity of 23.45°. Topography (including ice sheets and glaciers), bathymetry and the corresponding files are taken from the ETOPO2



**Table 1.** Models description: module names with corresponding number of vertical levels or type of coupling.

|  | *biogeodyn-MITgcmIS* | Levels/type | IPSL-CM6A-LR | Level/type | NorESM2-LM | Level/type |
|---|---|---|---|---|---|---|
| Resolution | 2.8 ° | NA | 1.6° | NA | 2° | NA |
| Land | LAND | 2 | ORCHIDEE | 11 | CLM5 | 15 |
| Atmosphere | SPEEDY | 5 | LMDZ6A-LR | 79 | CESM2.1-CAM6 | 7 |
| Ocean | MITgcm | 25 | NEMO-OPA | 75 | BLOM | 75 |
| Ice sheet | *MITgcmIS* | offline | NA | NA | NA | NA |
| Vegetation | BIOME4 | offline | ORCHIDEE | online | CLM5 | online |
| Sea ice | THSICE | online | NEMO-LIM3 | online | CICE | online |

dataset (ETOPO2, 2006) with a resolution of 2 arcminutes. Annual mean values of bare-surface albedos (in the absence of snow or sea ice) and fraction of land-surface covered by vegetation are the same as those used in Molteni (2003) and derived from the ERA dataset. A smoothing procedure is applied to convert these maps to the resolution of the MITgcm CS32 grid.

In order to assess the ability of *MITgcmIS* to correctly generate the ice sheets, we need to provide an input map of land elevation in the absence of ice. Such map was generated using the BedMachine dataset that is part of the MEaSUREs program of NASA (Morlighem, 2022; Morlighem et al., 2022), with the addition of an isostatic correction as in Paxman et al. (2022) for Antarctica and Greenland. The BedMachine dataset provides a density-corrected satellite-based DEM of the ice sheet surface, as well as a data-constrained estimate of bedrock elevation and a mask for identifying the different parts of the ice sheets.

There are two separate products for Antarctica (Morlighem, 2022) and Greenland (Morlighem et al., 2022), respectively. The surface elevation of the ice sheets are used to assess the performance of *MITgcmIS*, while bedrock elevations are used as initial boundary conditions. For both, we apply the same smoothing procedure as for ETOPO2, taking into account the different resolutions. The resulting isostatic map is shown in Fig. 2 on the CS32 grid. Note that, regarding paleoclimate simulations, PANALESIS or other reconstructions directly provide bedrock elevations.

Finally, it is important to also describe the tuning procedure. In order to obtain pre-industrial conditions at 280 ppm, once all the albedo values for vegetation cover, snow and ice have been fixed, we tune the relative humidity threshold for the formation of low clouds (a parameter denoted as $RHCL2$ in SPEEDY) so that the average global SAT becomes approximately equal to the observed value of 13.7°C (NOAA National Centers for Environmental Information, 2024). The adjusted value, $RHCL2 = 0.808$, is applied to all simulations. Moreover, the pre-industrial run was used to determine the coefficient $a$ in

Glen's law, which in our simulations is assumed to be constant (see eq. 2) and governs the ice sheet formation. This value has been set to $a = 1.2 \cdot 10^{-15}\,\mathrm{Pa}^{-3}\,\mathrm{s}^{-1}$ and its justification based on the SMB of Antarctica can be found in Sec. 3.2.4.

### 3.1.2 Comparison with data and CMIP models

To assess the performance of our climate framework, we run two simulations. The first one, denoted as run1, is the pre-industrial simulation at 280 ppm. This simulation will be assessed against two CMIP6 models, as there is a lack of observational data for





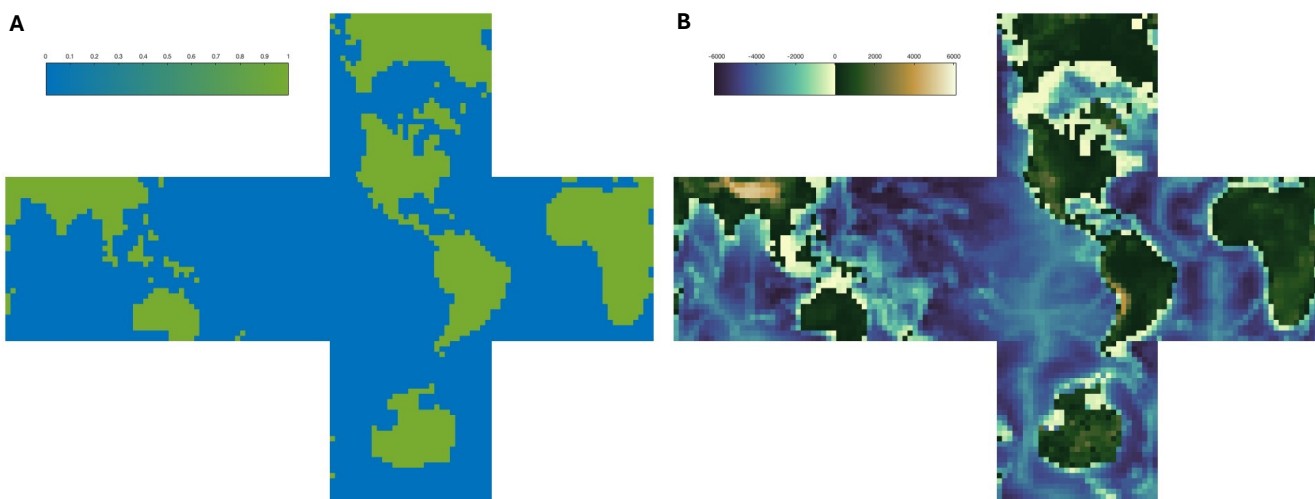

**Figure 2.** Initial conditions given to MITgcm on the cubed-sphere CS32 grid with (A) the mask file and (B) the bedrock topography and bathymetry.

this period. The second simulation (run2) corresponds to the 1979-2009 period, which has an average $CO_2$ concentration of 360 ppm (Lan et al., 2025). This run will be evaluated against the reanalysis and observational data.

More specifically, for assessing the pre-industrial run at 280 ppm, we use the output data from the IPSL-CM6A-LR model (Boucher et al., 2018) and the NorESM2-LM model (Seland et al., 2020) using the *piControl* dataset (Eyring et al., 2016). We chose these two CMIP models because they include dynamical vegetation. For the second run at 360 ppm, we compare

our data with the following datasets: ERA5 (Hersbach et al., 2023) for the atmosphere, OSRA5 (Copernicus Climate Change Service, 2021) for ocean diagnostics, MODIS for the vegetation, BedMachine for the surface elevation of Greenland and Antarctica (Morlighem et al., 2022; Morlighem, 2022), RAPID observations for the AMOC profile (Moat et al., 2025), and the Sea Ice index for the sea ice extent (Fetterer et al., 2002).

Two iterations of the procedure illustrated in Fig. 1 were necessary to reach convergence. Convergence is considered achieved

when less than 10% of the land points experience a change in biome distribution and total ice sheet volume. Moreover, the coupled MITgcm simulation needs to have a surface energy imbalance $F_s < 0.2\,\mathrm{W\ m^{-2}}$, corresponding to extremely low drifts in both the global ocean temperature and the SAT. The last 30 years of the second run are used for diagnostics. To be consistent in comparisons, all diagnostics are converted to the same longitude-latitude coordinate system with a spatial resolution of $2°$ $\times\ 2°$.

The outputs were treated using Matlab version 2023b and the figures were made using python. The MITgcm took approximately one week to reach equilibrium using 25 processors. BIOME4 and pysheds run in less than 5 minutes on a desktop computer. *MITgcmIS* needs 5 hours of CPU time due to the daily stepping of the ODEs in eqs. (6)-(7) for all land points.





**Table 2.** Global annual mean values averaged over the last 30 years, and associated standard deviations derived from interannual variability.

|  | run1: pre-industrial conditions | | | run2: 1979 - 2009 conditions | |
|---|---|---|---|---|---|
|  | IPSL-CM6A-LR | NorESM2-LM | *biogeodyn-MITgcmIS* | ERA5/OSRA5 | *biogeodyn-MITgcmIS* |
|  | 1850–1880 | 1850–1880 | 280 ppm | 1979–2009 | 360 ppm |
| SAT (°C) | 12.8±0.1 | 14.4±0.1 | 13.92±0.06 | 14.2±0.2 | 16.39±0.05 |
| $R_t$ (W m$^{-2}$) | 0.8±0.6 | 0.0±0.3 | -12.2±0.1 | 0.4±0.6 | -12.1±0.1 |
| $F_s$ (W m$^{-2}$) | 1.3±0.2 | 0.6±0.5 | 0.0±0.5 | 7±2 | 0.1±0.1 |
| NH ice extent (10$^6$ km$^2$) | 12.7±0.6 | 11.0±0.1 | 9.2±0.1 | 9.385±0.001 | 7.0±0.2 |
| SH ice extent (10$^6$ km$^2$) | 13.4±0.6 | 6.9±0.5 | 14.0±0.7 | 8.732±0.001 | 4.3±0.2 |
| $E - P$ (10$^{-8}$ kg m$^{-2}$ s$^{-1}$) | -7±2 | -3±2 | -2.3±0.9 | 0.08±0.01 | -2±1 |
| SST (°C) | 16.29±0.06 | 17.84±0.08 | 17.23±0.03 | 17.6±0.1 | 18.63±0.02 |
| SSS (psu) | 34.37±0.05 | 32.189±0.008 | 35.42±0.01 | 30.72±0.03 | 32.681±0.007 |

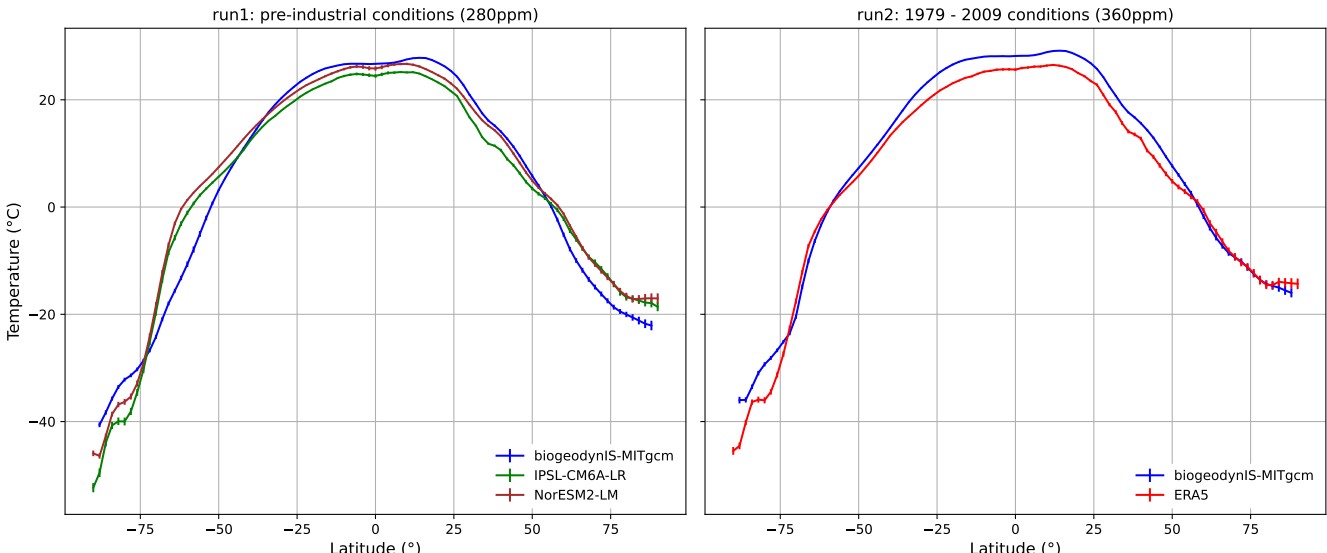

**Figure 3.** Zonal annual mean surface air temperature in °C for the two runs.

## 3.2 *biogeodyn-MITgcmIS* evaluation

To evaluate if our coupling setup correctly reproduces the modern Earth, we examined several diagnostics of the dynamical
behavior of atmosphere, ocean, vegetation and cryosphere.



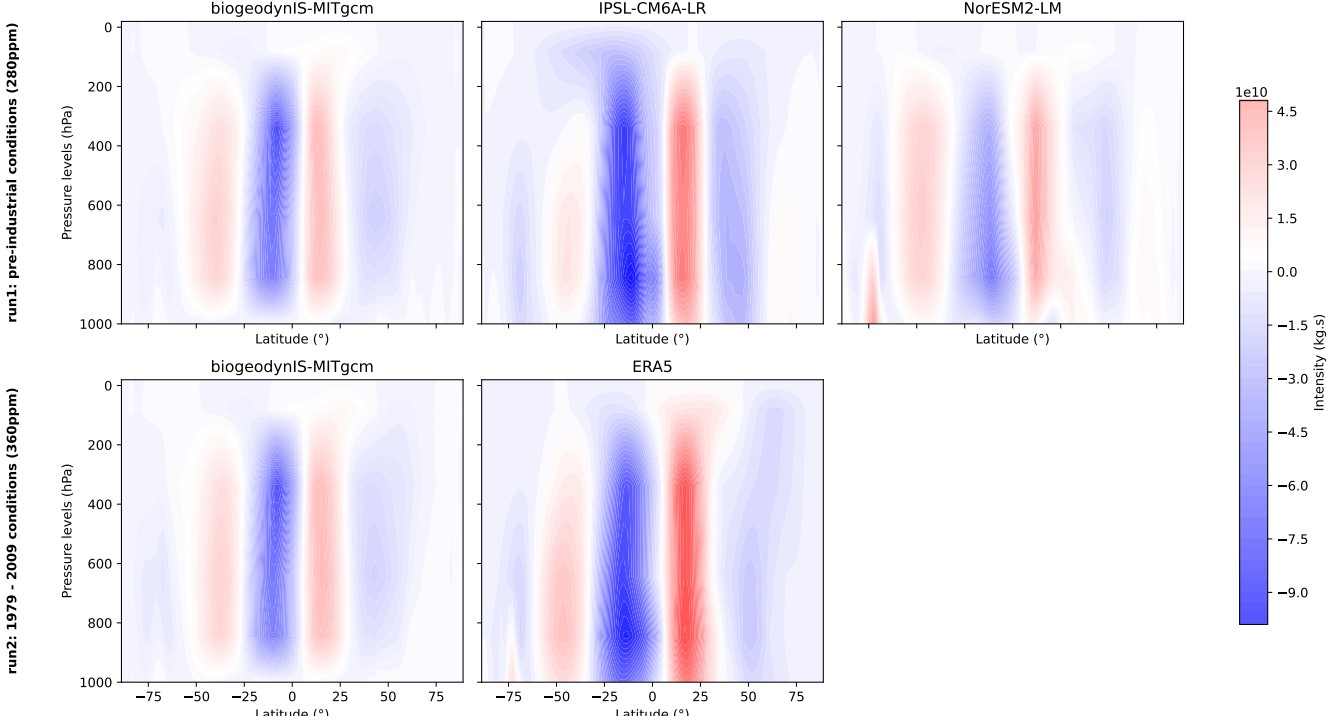

**Figure 4.** Atmospheric overturning cells for the two runs.

### 3.2.1 Atmosphere

In Table 2 global mean values of the relevant variables, calculated from the last 30 simulated years, are listed, together with the reanalysis data and the climatology values from the two CMIP models. The global mean SAT of 13.92 °C in run1 is intermediate between the two CMIP values, and close to the real data of 13.7 °C (NOAA National Centers for Environmental Information, 2024) because of our tuning procedure. However, the global mean SAT of 16.39 °C in run2 is larger than the ERA5 value. This corresponds to an Earth System Sensitivity (ESS) of 6.8°C, which is in the range of values obtained by other CMIP-class models, from 3 to 10 °C (Haywood et al., 2020). Excluding the ice sheet formation, the MITgcm coupled setup gives an Equilibrium Climate Sensitivity of approximately 4 °C, i.e., in the highest range of CMIP6 values (Nijsse et al., 2020; Zelinka et al., 2020).

More in detail, if we look at the SAT zonal profile as shown in Fig. 3, we can make common remarks for the two runs concerning the polar regions. The overall behaviour is well reproduced in our simulations, except the temperature at the south polar region, which is around 10°C higher in *biogeodyn-MITgcmIS* compared to the CMIP6 models and the ERA5 values, as also shown in Fig. A1. The difference in the north polar region is mainly located in Greenland. This can be due to the coarser resolution of the MITgcm coupled setup (2.8°), which underestimates the elevation of Antarctica and Greenland, hence giving higher temperatures than observations and CMIP models, where the ice sheet elevation is fixed to observed values.

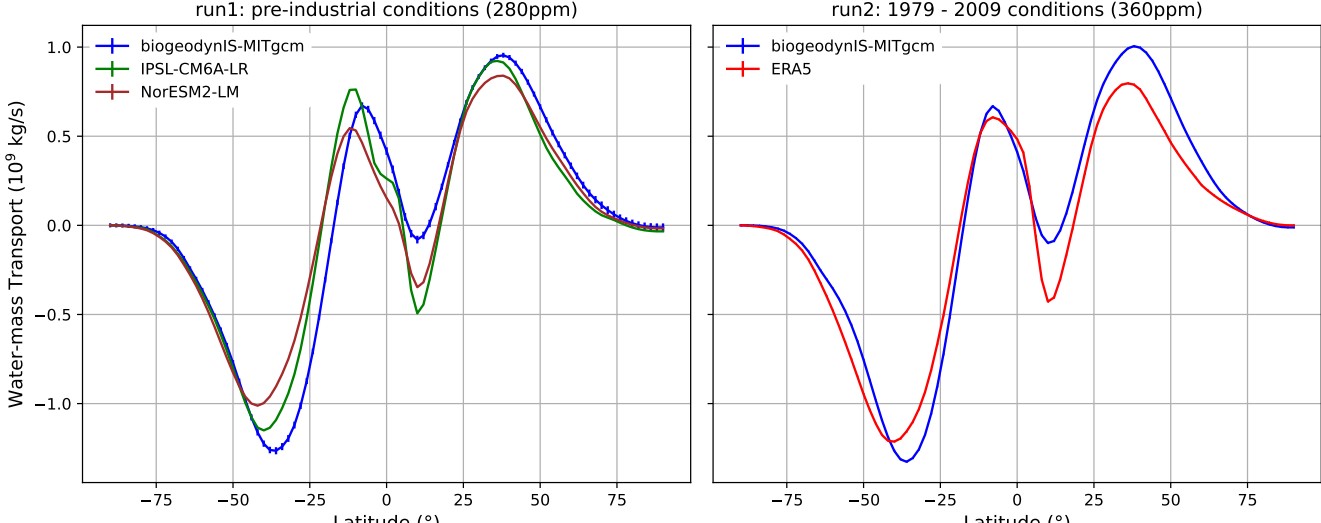

**Figure 5.** Northward water-mass transport in the atmosphere for the two runs. Data are plotted using the Carissimo correction (Carissimo et al., 1985).

Another important feature to be checked regarding the atmosphere dynamics is the model capability to correctly reproduce the Hadley cells. The comparison is shown in Figure 4. In run2 our model gives rise to a weaker positive overturning cell near the Equator, as also seen by Ruggieri et al. (2024) where the 8-layer SPEEDY module is coupled with the NEMO ocean. However, from run1 we see that the positive overturning cells reconstructed by NorESM2-LM are similarly weak, despite a
number of vertical layers in the atmosphere that is larger than the 5 layers in SPEEDY-MITgcm. In run1, SPEEDY gives cells with similar extent as the IPSL-CM6A-LR model, which has a state-of-the-art atmospheric module with 75 levels, while there is an additional positive south polar cell in the NorESM2-LM model. The lower branch of the positive Hadley cell is less intense than that in IPSL-CM6A-LR and ERA5. This feature has direct consequences on southward transport of water mass in the tropics, as shown in Fig. 5, which in our setup is indeed weaker than observations and CMIP models. In contrast, the
transport towards the southern polar region turns out to be larger in our simulations due to the comparatively intense Ferrel cell in the southern hemisphere. In addition, the mean zonal wind in our simulations, shown in Fig. A2, agrees with ERA5 and NorESM2-LM, with a slightly lower intensity of the jet stream in the northern hemisphere. Despite these limitations, it is important to note that the total water mass $E - P$ shows only a small imbalance in our simulations with respect to the control runs of the two CMIP models, as reported in Table 2.
As we can see in Fig. 6, while the precipitation peak at the ITCZ is correctly reproduced in our simulations, at a mean latitude of approximately 6°N (Marshall et al., 2014) that corresponds to the ascending branch of the Hadley cells, the precipitation intensity at ITCZ is underestimated due to weak Hadley overturning cells. In addition, our simulations do not capture the decrease in precipitation intensity at the Equator, as also observed in Ruggieri et al. (2024). The precipitation is in general overestimated in *biogeodyn-MITgcmIS* with respect to observations in subtropics and extratropics. However, the SPEEDY



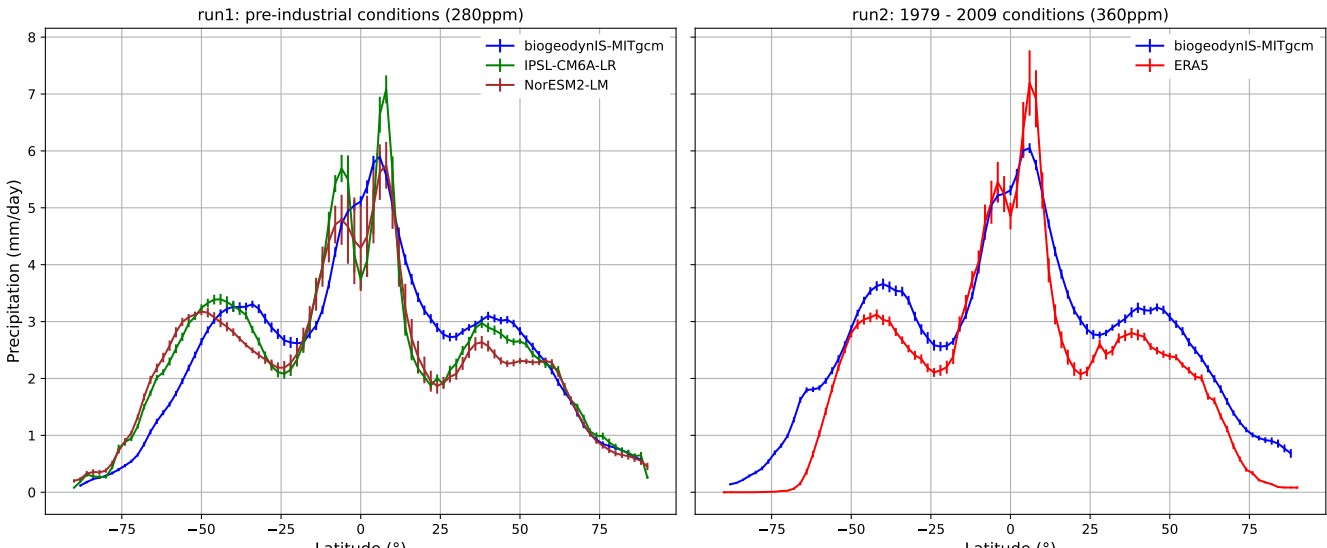

**Figure 6.** Zonal annual mean precipitation in mm/day for the two runs.

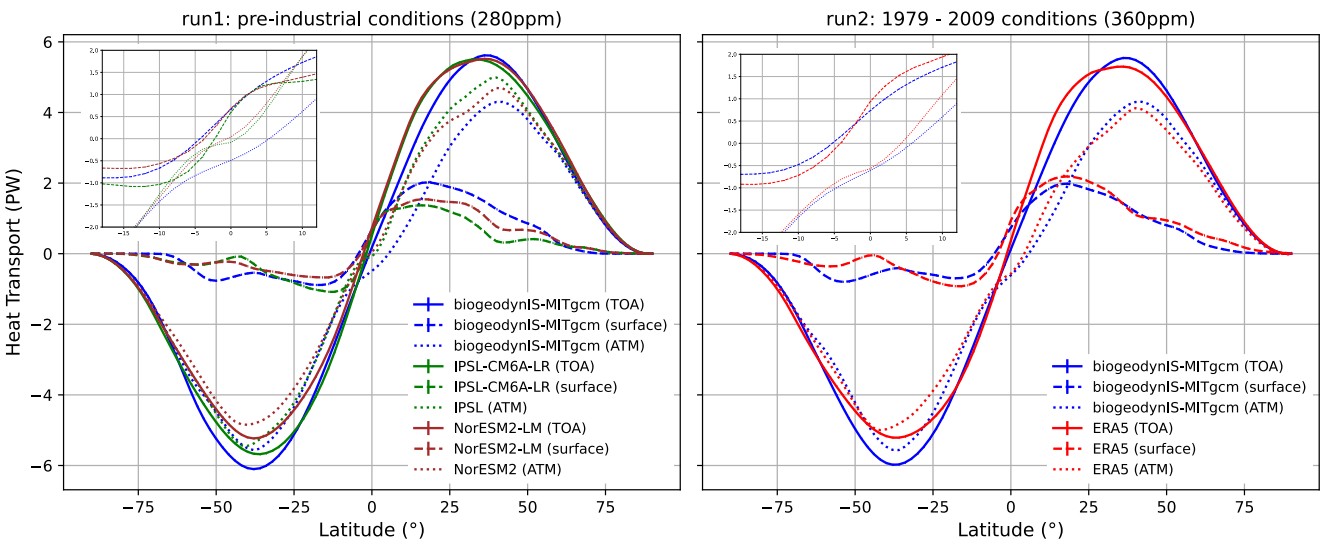

**Figure 7.** Northward heat transport at the top of the atmosphere (solid lines) and at the surface (dashed lines). Their difference gives the heat transport in the atmosphere (dotted lines). The insets show a zoom of the tropical region.

module captures the overall precipitation pattern in both runs (see Fig. A3), with localised maximum anomalies of around 5 mm/day in the equatorial region.





The heat transport at TOA in Fig. 7 shows that *biogeodyn-MITgcmIS* closely follows the overall pattern in both runs, except for a sligthly stronger southward heat transport at approximately 40°S. Across the Equator, the atmospheric heat transport (dotted lines) is southward in our simulations and in ERA5, correctly compensating for the northward ocean heat transport

driven by the AMOC, as described in Marshall et al. (2014). In contrast, the CMIP models do not show this compensation across the Equator despite their finer vertical resolution in the atmosphere, as can be seen in the inset of Fig. 7. The *biogeodyn-MITgcmIS* coupled system has not a closed energy budget $R_t$ at the top of the atmosphere (TOA), as shown in Table 2. This is mainly due to the fact that the energy used for ice sheet growth is not included in this diagnostics. As further confirmation, during the first MITgcm iteration before running the ice sheet model, $R_t$ is indeed comparable to the values obtained by the

CMIP models and ERA5 (see also other present-day simulations performed with the MITgcm coupled setup in Brunetti and Vérard (2018)).

### 3.2.2 Ocean

In order to assess the capacity of *biogeodyn-MITgcmIS* to correctly represent the ocean dynamics, we looked at the sea surface temperature (SST) and salinity (SSS), the sea ice extent, the water-mass budget, the AMOC profile and the heat transport at

the surface.

As shown in Table 2, SST in run1 of *biogeodyn-MITgcmIS* simulations is in between the two CMIP models, whereas it is higher than in OSRA5 being consistent with SAT. Sea ice extent in our run1 simulation is lower in the northern hemisphere (NH) than in the southern hemisphere (SH), in contrast to NorESM2-LM results. Note that the NorESM2-LM model simulates a pre-industrial climate with less SH sea ice (with an extent of $6.9 \cdot 10^6$ km$^2$) compared to ERA5 (with approximately $8.7 \cdot$

$10^6$ km$^2$), even though ERA5 reflects a climate state with a higher atmospheric $CO_2$ content.

For run2, the values of sea ice extent in *biogeodyn-MITgcmIS* are lower than those in ERA5, especially in the southern hemisphere, due to higher surface temperatures. We observe that sea ice extent obtained in our simulations, which is shown in Fig. A4, is in good agreement with that in Ruggieri et al. (2024), obtained with SPEEDY-NEMO for the period 1979-2014.

Our two simulations show a reduction in sea ice extent with increasing atmospheric $CO_2$ concentrations, along with a

consistent decrease in salinity due to enhanced water dilution. The total extent of sea ice in our run1 simulation is larger than in NorESM2-LM, which explains a higher value of salt concentration. It is comparable to that of IPSL-CM6A-LR, which, however, shows an imbalance in the water budget of $E - P = -7 \cdot 10^{-8}$ kg m$^{-2}$ s$^{-1}$ and a slightly lower value of salinity.

The ocean heat transport (OHT) in Fig. 7 (dashed lines) shows a larger amount of heat towards the northern polar region compared to CMIP models, explaining why our setup produces less sea ice there. The bulk of the OHT is dominated by

the Ekman transport in the subtropical gyres. As mentioned before, the AMOC effect is to increase heat transport across the Equator (Marshall et al., 2014), which is of around 0.7 PW in all models. It is important to note that the surface energy imbalance $F_s$ in our simulations is very low ($|F_s| \leq 0.1$ W/m$^2$, see Table 2), because they are run close to equilibrium. In contrast, the observations give $F_s = 7$ W/m$^2$, reflecting forced conditions. Although they represent control runs, the two CMIP simulations exhibit values larger than 0.1 W/m$^2$, indicating that they are not fully equilibrated.





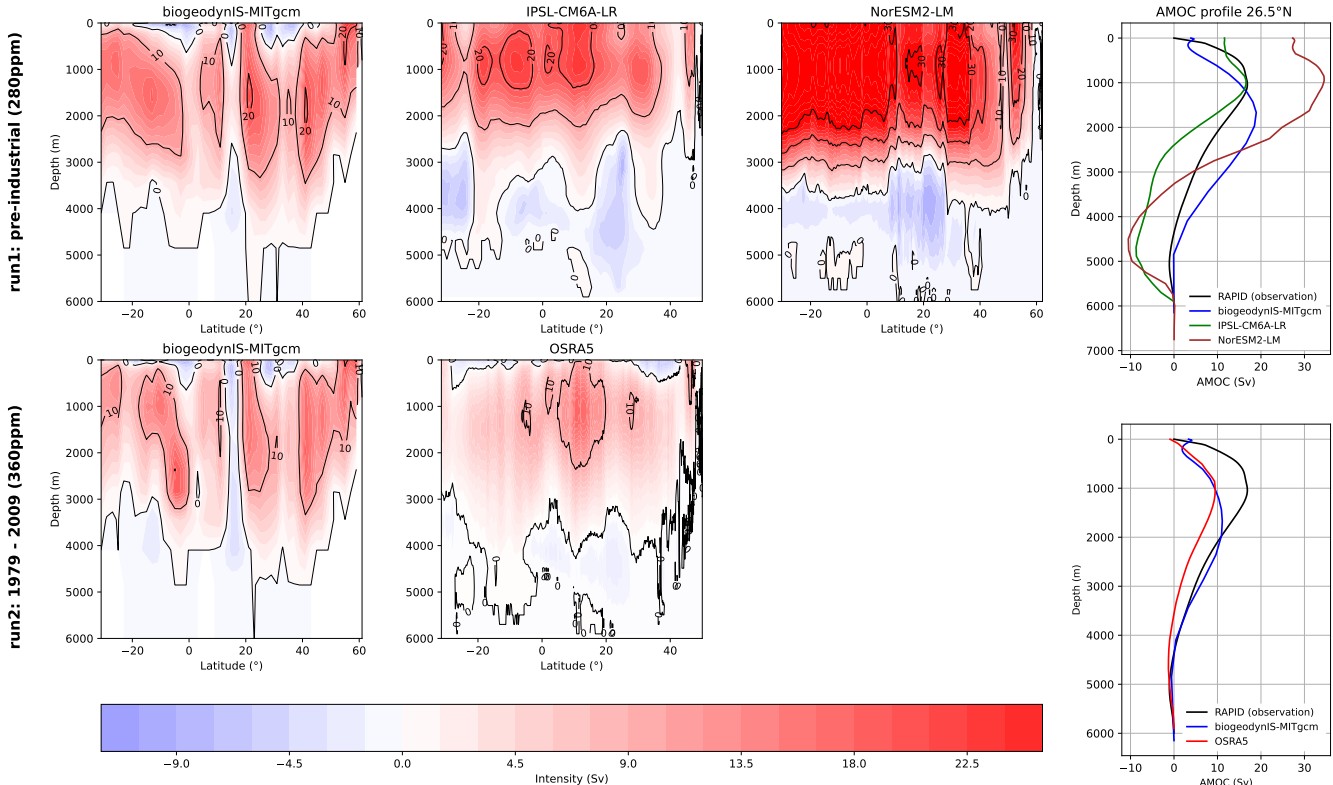

**Figure 8.** AMOC intensity for the two *biogeodyn-MITgcmIS* simulations, CMIP models and OSRA5. The vertical profile at 26.5°N of the AMOC is added on the right with the addition of the RAPID observations.

As shown in Fig. 8, the AMOC produced by our coupled setup for run1 has a clockwise (positive) overturning cell with intensity comparable to the cell obtained by IPSL-CM6A-LR, which however develops at lower depths. The positive overturning cell in NorESM2-LM has a higher intensity than both *biogedyn-MITgcmIS* and IPSL-CM6A-LR. For run2, the AMOC intensity produced by OSRA5 exhibits a maximum at the Tropics, while in our coupled setup there are several regions of high intensity. In both, there is a weak anticlockwise (negative) overturning cell at depths higher than 4 km. Right panels in Fig. 8

show vertical profiles at 26.5°N of the AMOC streamfunction. The two blue curves correspond to the *biogeodyn-MITgcmIS* simulations and show that there is a decrease in the intensity of the AMOC positive cell as the $CO_2$ concentration increases. This behavior is expected and demonstrates the ability of *biogeodyn-MITgcmIS* to produce consistent results. It is important to note that the two CMIP models exhibit strong negative values around 4500 m that are not appearing in the observations. A similar pattern is also observed in another class of CMIP models described in Valdes et al. (2017), even in present-day simu-

lations. However, *biogeodyn-MITgcmIS* shows the maximum of the positive cell around 1500 m, whereas it should be around 1000 m according to the RAPID measurements. This discrepancy may be related to the fact that the MITgcm ocean module includes only 25 vertical levels, compared to 75 in the two CMIP models.





### 3.2.3 Vegetation

In this section, we evaluate the capacity of the coupled system to correctly reproduce the present-day vegetation and to assess
its performance against models with dynamical vegetation.

As shown in Fig. 9, the *biogeodyn-MITgcmIS* simulations give rise to a good representation of the major biomes. They both display the boreal forest, which, following the biome classification used in Kaplan et al. (2003) and Haywood et al. (2010), corresponds to cool mixed forests, evergreen and deciduous taiga (see the legend at the bottom of Fig. 9). They also display the Amazon rainforest by returning the tropical evergreen forest biome, which has a larger extent for higher $CO_2$ values. Both
simulations include the desert biome, whose extent reduces with increased atmospheric $CO_2$ content, although smaller than expected. This is directly linked to the excess of precipitation produced by the SPEEDY module at these latitudes (Fig. 6). Major differences between run1 and run2 are the emergence of tropical savanna in South America and Africa in run2, the loss in run2 of shrub tundra at high latitudes, which has been replaced by deciduous taiga, temperate grassland or montane forest. In Greenland, run2 shows an increase in cushion-forb and lichen, and the appearance of prostrate shrub tundra, which was not
present in run1. Run2 shows an expansion of temperate grassland across North America and Russia.

The zonal profiles of the Net Primary Production (NPP) distribution are shown in the right panels of Fig. 9. Our simulations reproduce the general pattern by correctly displaying an increase in the equatorial region. However, they fail to capture the decrease at $10°$ and $-25°$ latitude, as shown by MODIS. It is important to note that there is a difference in NPP intensity between run1 and run2. This is mainly explained by the increase of $CO_2$ concentration, precipitation and temperature, as
shown in Figs. 3 and 6. The two CMIP models show different behaviors. NorESM2-LM captures quite well the decrease at $10°$ and $-25°$, as well as the increase in the equatorial region, despite lower intensities than MODIS. IPSL-CM6A-LR does not show vegetation at latitudes higher than $25°N$, as ORCHIDEE does not include high-latitude biomes such as tundra (Dinh et al., 2024). The two dynamical vegetation models do not capture all the trends in the NPP pattern, even with more sophisticated land modules than the one used in our setup. This is confirmed by the Pearson correlation coefficient and slope values obtained
in our two simulations and the two CMIP models when assessed against the MODIS observations, as shown in Fig. A5. Both *biogeodyn-MITgcmIS* runs are evaluated against MODIS due to the lack of observational data in the pre-industrial era. For both runs of our setup, the correlation coefficient $r$ and slope values are close to one, with an increase in slope in run2. In contrast, the two CMIP models exhibit lower slope values, and IPSL-CM6A-LR also shows a lower $r$ value. This emphasizes that, even without online dynamical vegetation, our setup successfully captures global-scale vegetation features and remains consistent
with different climate conditions.

### 3.2.4 Ice sheet

The performance of *MITgcmIS* is evaluated against present-day observations using the BedMachine datasets. Since there are no observational data available for the pre-industrial period, and the two CMIP models do not include dynamical ice sheets, both *biogeodyn-MITgcmIS* simulations are assessed using the same set of present-day observations. As mentioned in Sec. 3.1.1,
evaluating the SMB produced for Antarctica is a prerequisite to calibrate the Glen's law parameter using the total ice sheet



**Figure 9.** Biome map for the two *biogeodyn-MITgcmIS* simulations, and zonal NPP profile (right panels) for all models and MODIS dataset.





**Figure 10.** Map of the region where ice sheets form in the two *biogeodyn-MITgcmIS* runs, showing the surface elevation of both ice-free areas and ice sheets.

volume in our setup. The reason is that the volume is strongly sensitive to both net SMB and the Glen's law coefficient; without a good estimate of SMB, the correct volume can be achieved for the wrong reasons. Thus, we calibrated the Glen's law parameter based on the first iteration of run1. To this end, we tested a range of $a$ values that yield different Antarctic volumes (Tab. A1), while maintaining a similar surface elevation profile. We selected $a = 1.2 \cdot 10^{-15}$ Pa$^{-3}$s$^{-1}$ as a compromise, as it

produces a volume within 10% of the observed value and surface elevations consistent with observations. For the simulation at 280 ppm (run1), we obtained an SMB of approximately 1500 Gt yr$^{-1}$. This SMB value can be compared with the ensemble mean of SMB values for Antarctica obtained from a comparison of Regional Climate Models (RCMs) in Mottram et al. (2021). In that study, the ensemble mean over the grounded ice sheet is estimated at $2073 \pm 306$ Gt/yr$^{-1}$. Although the value obtained



**Table 3.** Ice volume formed in the two *MITgcmIS* runs.

|  | 280 ppm | 360 ppm |
| --- | --- | --- |
| Antarctica volume ($10^6$ km$^3$) | 21.6 | 21.0 |
| Greenland volume ($10^6$ km$^3$) | 2.6 | 1.5 |

in our simulation is slightly lower than the ensemble mean reported in Mottram et al. (2021), as well as lower than the values

obtained by all individual models in that comparison, it is important to consider that the spatial smoothing applied to obtain 2.8° resolution in our simulations implies a different representation of the Antarctic continent compared to models with higher resolutions (25-50 km). Additionally, there are known simplifications in the description of the ice sheet in *MITgcmIS*, as discussed in Sec. 2.4, and limitations in the representation of snow processes in the land module. Therefore, even if our value is slightly lower than those obtained from RCMs, it remains within the same order of magnitude.

The ice sheets obtained in both runs are shown in Fig. 10. The Antarctica ice sheet does not change significantly from one run to the other, in both height and extent. However, the Greenland ice sheet decreases strongly in height and extent when the $CO_2$ content increases. A total volume of $24.2 \cdot 10^6$ km$^3$ is formed in run1, compared to $22.5 \cdot 10^6$ km$^3$ in run2 (Tab. 3). For Antarctica, there is a difference of 6% for run1 and 8% for run2 compared to the observed volume (after smoothing to the same spatial resolution) of $22.9 \cdot 10^6$ km$^3$. Hence, in both runs the ice sheet volume is of the same order as the observations, with

a decrease in run2 that corresponds to a total freshwater flux in 145 years of approximately 0.37 Sv, with a rate in Greenland of $1.6 \cdot 10^{-3}$ Sv yr$^{-1}$, and nearly two times smaller in Antarctica. These values are much smaller than those typically used in hosing experiments (e.g., 0.1-0.5 Sv yr$^{-1}$ in Jackson et al. (2023)), but they overestimate the melting rates reconstructed by a comprehensive survey in Greeland by one order of magnitude (Mouginot et al., 2019).

This overall agreement with observations is reflected in a statistically significant Pearson correlation coefficient $r$ of 0.84/0.81

in surface elevation between the ice sheets formed in *MITgcmIS* run1/run2 and BedMachine, respectively, as shown in Fig. 11, and detailed for Antarctica in Fig. A6. However, the slope values of 0.44 and 0.39 for run1 and run2, respectively, are lower than 1. This means that the model tends to underestimate the largest ice sheet heights and to overestimate the smallest ones on the edges. This pattern is evident in Fig. A7, particularly in West Antarctica. In contrast, conclusions about Greenland are more uncertain due to the limited pixel coverage. Overall, we observe that Antarctica in *MITgcmIS* agrees more closely with

observations than Greenland, which shows a strong sensitivity to an increase of atmospheric $CO_2$ content.

If we examine the histograms in the right panels of Fig. A7, we observe that *MITgcmIS* can reproduce both low and high ice sheet elevations over Greenland, albeit for a low number of grid points. For Antarctica, the histograms support the conclusions discussed above. The excess accumulation in West Antarctica is also reported in Xie et al. (2022), which uses an ice sheet model of similar complexity. In addition, analogous biases in ice thickness are also observed in more sophisticated models,

such as in Quiquet et al. (2018), which underestimates the ice thickness in central Antarctica of around 300-400 m. The fact that *MITgcmIS* does not correctly capture the peak of ice sheet elevation in central Antarctica can be attributed to the model's coarse spatial resolution, confirming the role played by spatial resolution in ice sheet models (Rückamp et al., 2020). In addition, it



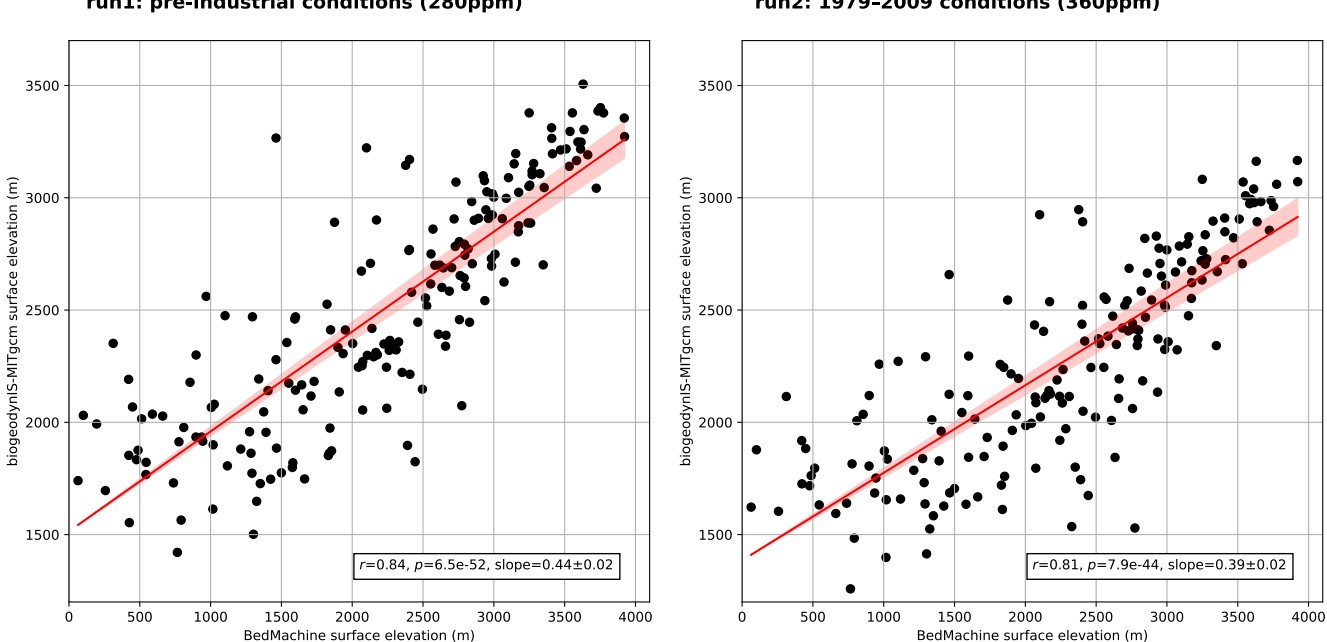

**Figure 11.** Correlations in surface elevation between the ice sheets formed in the two *MITgcmIS* simulations and BedMachine.

is important to consider uncertainties related to bedrock elevation, isostatic adjustment, and measurements of the ice sheet thickness. Nevertheless, we can conclude that our model reproduces the first-order characteristics of the ice sheets reasonably

well, and shows a consistent behavior between run1 and run2.

### 3.3 Future developments

In this paper, we have described how the implemented offline coupling framework successfully reproduces the large-scale climate and its major components, giving results comparable to those of two CMIP6 models. However, further improvements are possible.

The atmospheric module is currently based on a previous version of SPEEDY with five vertical levels, while a newer version with eight levels is now available (Kucharski et al., 2013). This updated version has recently been coupled with the NEMO ocean model (Ruggieri et al., 2024), providing a good representation of both climatology and the main modes of internal variability. A further improvement would be to directly implement the offline coupling with BIOME4 on the MITgcm cubed-sphere grid, thereby eliminating interpolation errors, as discussed in Sec. 2.6.

Regarding the ice sheet model, we plan to provide the option of performing both online and offline coupling with the MITgcm dynamical kernel. This upgrade requires an enhancement of the land module within MITgcm. While incorporating a detailed land surface scheme, such as the one used in the JULES model (Wiltshire et al., 2020) would constitute a major improvement, we plan to follow a different approach, as only selected processes need to be represented at coarse spatial resolutions. As





mentioned in the Methods section, the current two-layer land module in MITgcm lacks representations of processes such as heat conduction in snow, meltwater refreezing and retention, and snow compaction. All of these can significantly affect the ablation and accumulation in the surface mass balance, which is currently underestimated in our simulations. Using the open-source snowpack model described in Essery (2015) could be a viable option. In future iterations of *MITgcmIS*, sliding and basal heat balance could easily be implemented – this would allow study of nonlinear processes that occur over continental and millennial scales, such as binge-purge oscillations (MacAyeal, 1993). Furthermore, incorporating a dynamical ice sheet model would lead to a more consistent energy budget across the different model components. However, due to the slow temporal evolution of ice sheets starting from bedrock conditions, a spin-up phase using offline coupling will always be necessary, since the coupled MITgcm cannot be run on timescales of hundreds of thousands of years due to computational costs.

## 4    Conclusions

In summary, the *biogeodyn-MITgcmIS* coupled setup provides a good representation of large-scale features of the present-day climate with reasonably low computational costs. Atmosphere and ocean dynamics agree well with observations, giving a model performance comparable to CMIP6-class models. Coupling with the BIOME4 vegetation model reproduces the main biomes, with results similar to those obtained using CMIP6 models with dynamical vegetation. Moreover, the vegetation cover obtained with *biogeodyn-MITgcmIS* exhibits coherent behavior under increasing $CO_2$ concentrations. The ice sheet component, *MITgcmIS*, reproduces reasonably well the surface mass balance, as well as the global volume and the thickness of Antarctica and Greenland ice sheets, considering its coarse spatial resolution. An upgrade of the land module and the development of an online ice sheet module could address some of the limitations of the current version and are planned for future development.

For now, this new tool, which consistently describes the global-scale dynamics of the ocean, atmosphere, vegetation, and ice over multimillennial timescales with relatively low computational costs, allows for a new range of climate investigations. Tipping elements (McKay et al., 2022), like the Atlantic Overturning Meridional Circulation, ice sheets, Amazon rainforest, and sea ice extent, can be studied with a modelling framework that allows for the consistent evolution of all these interacting components at the global scale. An additional advantage is that *biogeodyn-MITgcmIS* is adaptable to different modelling setups, as each module can be removed if needed. Furthermore, by changing the coupling frequency, both equilibrium and transient simulations can be performed. We expect that the proposed model will contribute to the investigation of the climate system on Earth, for both present-day and past continental configurations, as well as on idealised scenarios and exoplanet research.

*Code and data availability.* The BedMachine data for Antarctica and Greenland can be accessed through the NASA National Snow and Ice Data Center at https://nsidc.org/data/explore-data. Data for the isostatic correction are available from the U.S. National Science Foundation Arctic Data Center at https://arcticdata.io/catalog/view/doi:10.18739/A2280509Z. The sea ice extent data can be downloaded from the National Snow and Ice data Center Sea Ice Index at https://nsidc.org/data/g02135/versions/3#anchor-data-access-tools. The MODIS TERRA data for the NPP can be obtained at https://modis.gsfc.nasa.gov/data/dataprod/. The RAPID data from the RAPID/MOCHA/WBTS project are available from https://rapid.ac.uk/. CMIP6 model data can be freely downloaded on the ESGF nodes (for example https://



esgf-node.ipsl.upmc.fr/search/cmip6). ERA5 and OSRA5 datasets are accessible via the Copernicus Climate Data Store at the following link: https://cds.climate.copernicus.eu/datasets. MITgcm is open source and archived on https://github.com/MITgcm/MITgcm, the vegetation model BIOME4 is available from https://github.com/jedokaplan/BIOME4, pysheds from https://github.com/mdbartos/pysheds.

The current version of *biogeodyn-MITgcmIS* is available from the project website https://doi.org/10.5281/zenodo.15584981 under the license Creative Commons Attribution 4.0 International. The exact version of the model used to produce the results used in this paper is archived on https://doi.org/10.5281/zenodo.15584981 under DOI:10.5281/zenodo.15584981 (Moinat et al., 2025), as are input data and scripts to run the model and produce the plots for all the simulations presented in this paper (Moinat et al., 2025).

*Author contributions.* MB planned the study and acquired funding. DNG implemented the main part of the *MITgcmIS* code with support from LM. FF and CV implemented the hydrology component. LM ran the simulations with the help of MB, and made the plots. LM and MB analysed the simulation results. LM, MB and DNG wrote the manuscript. All authors reviewed the paper.

*Competing interests.* The authors declare that they have no conflict of interest.

*Acknowledgements.* LM, FF, CV and MB acknowledge the financial support from the Swiss National Science Foundation (Sinergia Project No. CRSII5_213539), and useful discussions with the Sinergia team. LM, FF, CV and MB thank the pan-EUROpean BIoGeodynamics network (EUROBIG) COST Action (CA23150, https://www.cost.eu/actions/CA23150) and, in particular, Taras Gerya for inspiring discussions on biogeodynamics. LM acknowledges the financial support from the EUROBIG COST Action (CA23150). DNG acknowledges support from the Natural Environment Research Council (Project Nos. NE/X005194/1, NE/X01536X/1).





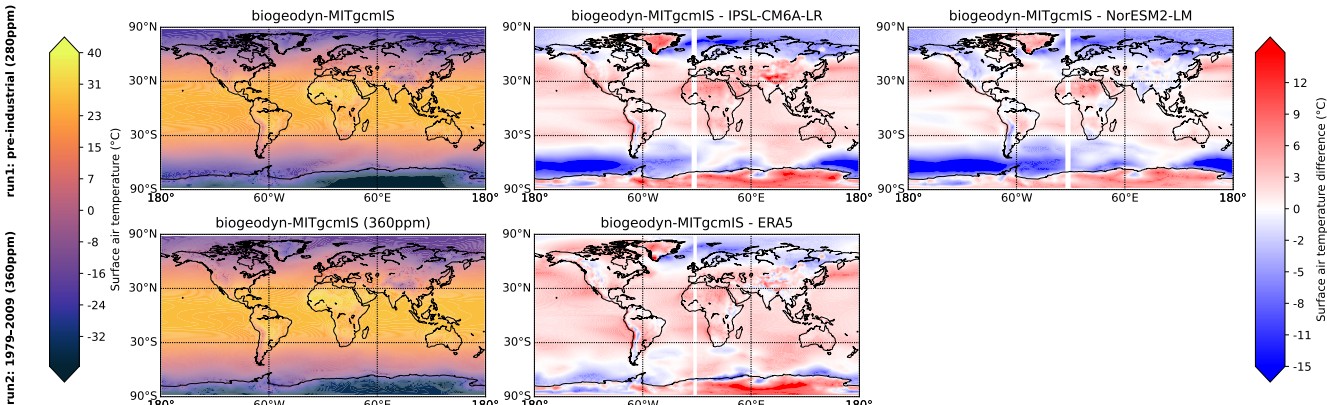

**Figure A1.** Climatology of surface air temperature for the two *biogeodyn-MITgcmIS* simulations and the corresponding anomaly maps with respect to CMIP models and ERA5.

## Appendix A: Appendix A

Additional figures, mentioned in the main text, are shown in this Appendix.





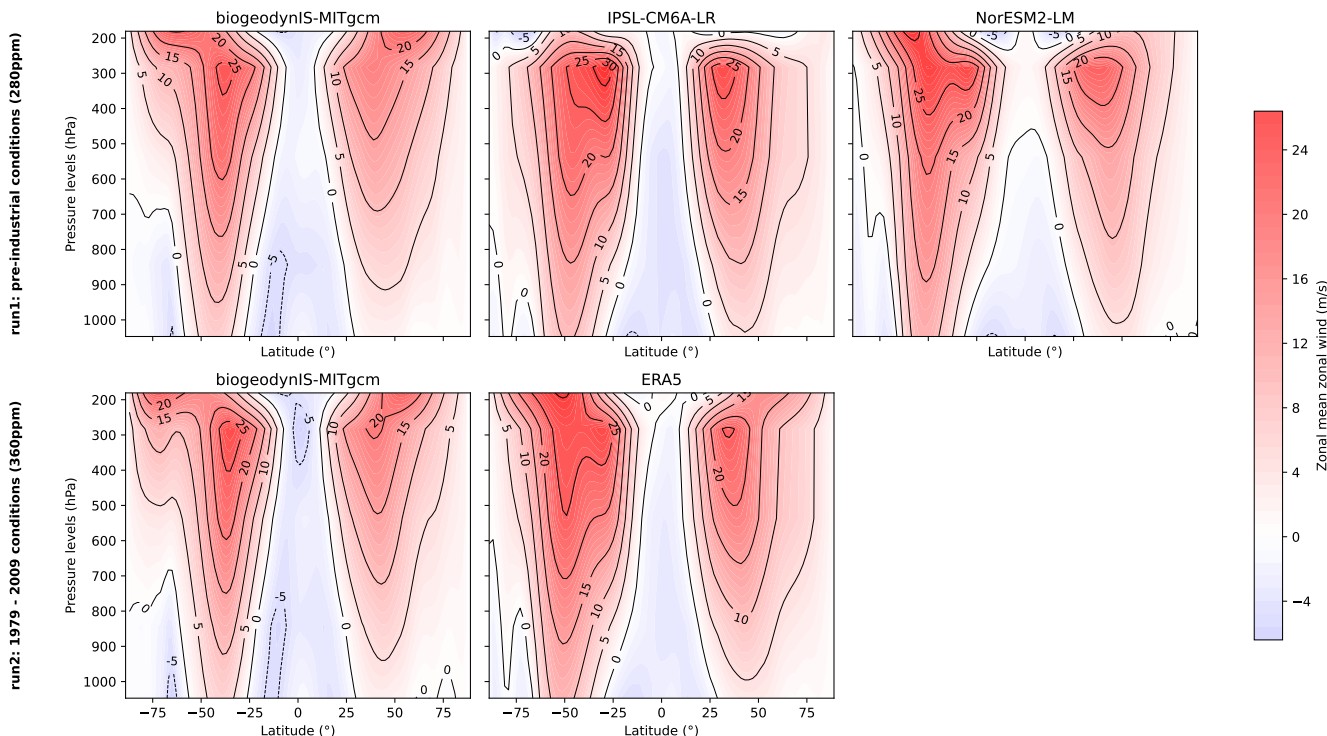

**Figure A2.** Zonal annual mean of zonal wind for the two runs.



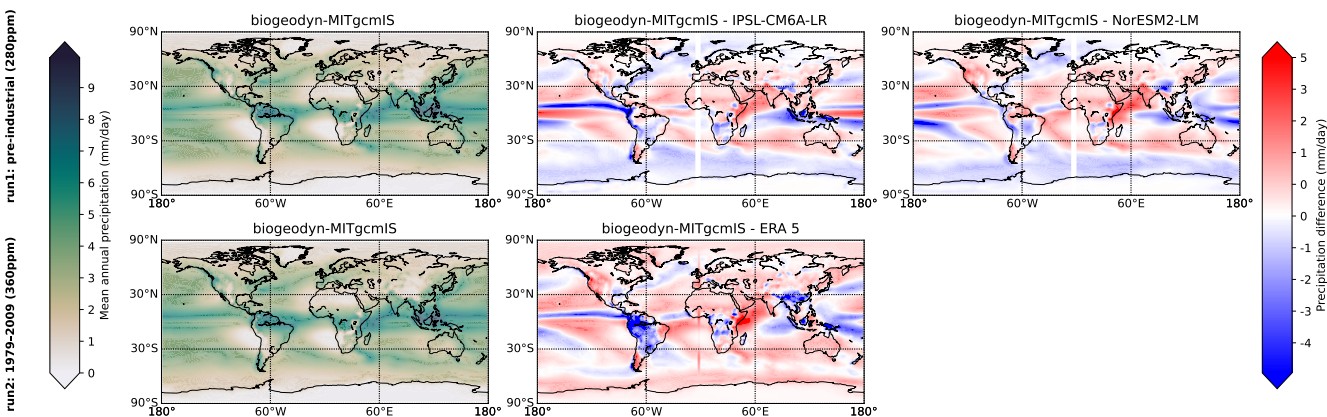

**Figure A3.** Climatology of precipitation for the two *biogeodyn-MITgcmIS* simulations and the corresponding anomaly maps with respect to CMIP models and ERA5.



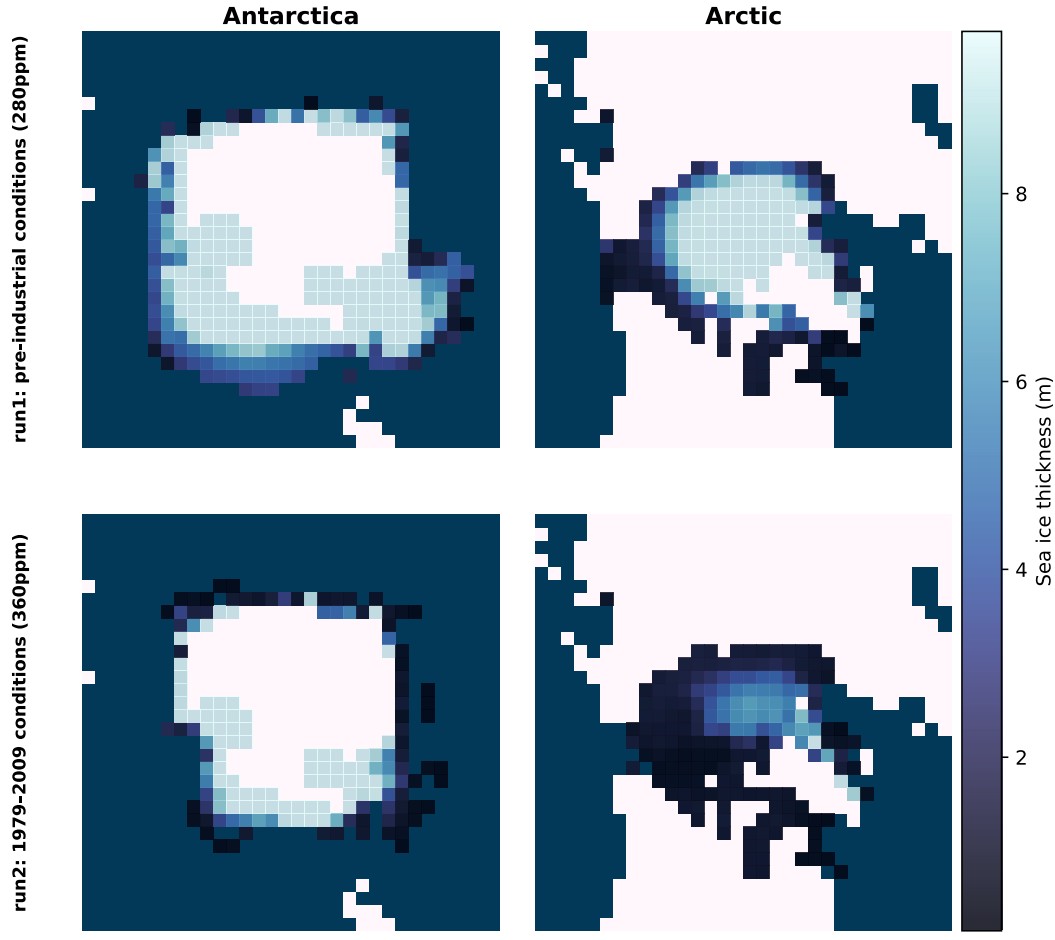

**Figure A4.** Annual mean sea ice thickness and extent in the polar regions for the two *biogeodyn-MITgcmIS* simulations.



**Figure A5.** Linear regression (with corresponding $r$, $p$ and slope values) of NPP obtained with each of the CMIP models and the two *biogeodyn-MITgcmIS* simulations with respect to MODIS dataset.





**Table A1.** Glen's law parameter and corresponding ice sheet volume produced in Antarctica by *MITgcmIS*.

| Glen's parameter ($10^{-15}$ Pa$^{-3}$s$^{-1}$) | Antarctica volume ($10^6$ km$^3$) |
|---|---|
| 0.6 | 23.5 |
| 1.2 | 21.6 |
| 1.7 | 20.7 |
| 2.5 | 19.0 |
| 3.2 | 18.8 |



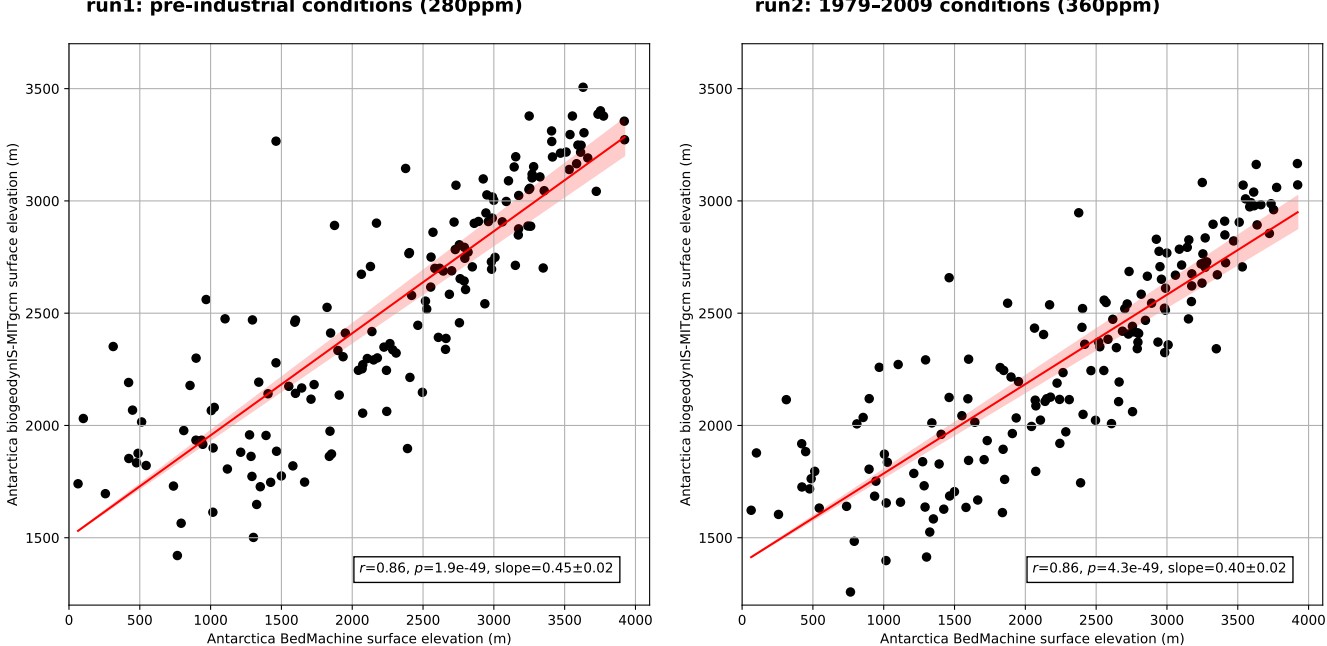

**Figure A6.** Correlations in surface elevation between the Antarctic ice sheet formed in the two *MITgcmIS* simulations and BedMachine.



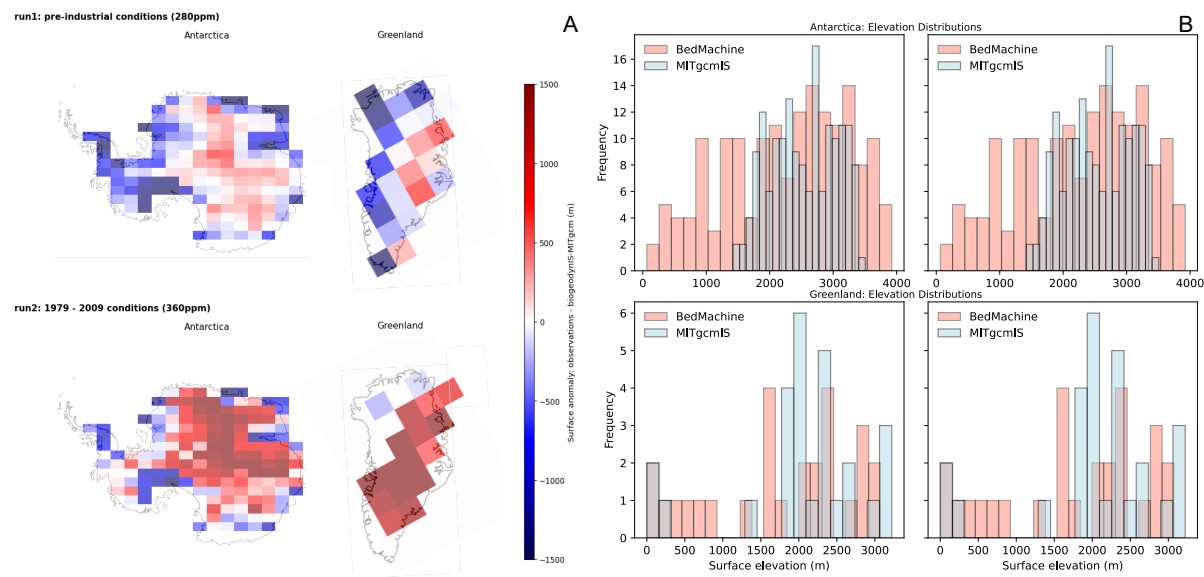

**Figure A7.** A. Anomaly maps of surface elevation (BedMachine minus the *MITgcmIS* simulations where ice sheets form). B. Histograms of surface elevation where ice sheets form in *MITgcmIS* simulations (blue) and in the BedMachine observations (orange).



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
