# Peer review of "biogeodyn-MITgcmIS (v1): a biogeodynamical tool for exploratory climate modelling"

_EGUsphere, 2025_

## Referee Comment (RC2)

**Moinat et al. – biogeodyn-MITgcmIS (v1): a biogeodynamical tool for exploratory climate modelling.**

**Summary**

Moinat and colleagues present the "biogeodyn-MITgcmIS", a global climate system model that is designed to encompass relatively low computational costs and yet couple extended components of the earth system – ocean, atmosphere, land surface, sea-ice, and ice-sheets. They do so by coarsening the grid of the MITgcm and, in regards to the land surface vegetation and ice-sheet models, by performing offline equilibration runs using the outputs of the MITgcm. This asynchronous running minimises computation costs that would otherwise be required in multimillennial simulations. Their procedure is to run the MITGcm, followed by offline runs of the ice-sheet model, water shed runoff model and vegetation model in that order.

The authors run two experiments: pre-industrial and a modern day simulations. They compare the preindustrial run to two CMIP6 models and they compare their modern day run against a suite of observational products. The model shows broad agreement against these datasets in this assessment.

The work is presented in a clear manner and the figures and writing are of good quality. The authors do not over-reach and the paper is appropriate for the journal.

My only major comment is that the authors talk about climate tipping points in the introduction and conclusion, and yet, the authors and the experiments they have performed have not convinced me that this model is indeed able to simulate a climate tipping point where the system rapidly shifts to a new attractor. I challenge the authors to provide such an experiment and talk to this within a new section of their results. Given the low computational costs of their model, the authors could achieve this by performing a 4x CO2 experiment, for instance, both with and without their ice-sheet model and then assessing the impact of including that model on the system. Or perhaps the authors have better ideas about this sort of feedback response could be achieved?

Minor comments:

Line 33: "are fast reaching a stationary state" → "can reach a stationary state rapidly"

Line 44: "such as **the** deep ocean"

Line 47: "However, high computational costs makes these models appropriate for"

Line 59: I think it's best to remove all future tenses from the paper. Please make your writing either present tense or past tense.

Line 85: "to capture mixing by mesoscale…"

Line 101: "will be" → "is"

Lines 100 – 106: Are all these features done automatically by the model or do you have to do these manually?

Line 114: Define what a PFT is. Also, make sure to correct your acronyms to "PFT", not "PTF".

Line 133: How is this different to your new model? Do you use these models together in the same runs?

Line 135: Can you spell out SMB instead of adding a new acronym? You already have a lot of acronyms.

Line 169: It's interesting to me that you assign "a" as constant when making it vary with temperature or something else would actually provide your model with the kind of feedback that would create non-linear tipping points in the system.

Line 217: What is $w_{ss}$. Please define.

Line 267: How many models actually do this amongst the CMIP6 contingent?

Line 284: "corrections"

Line 292: "and giving a new runoff map as output".

Fig 1: Can you add numerals "1" and "5" to your steps in this figure? You have "2", "3" and "4".

Line 316: "This map was…"

Table 2: Please make the caption more informative. Tell me what all variable are.

Line 372: "… reproduce Hadley cells (Figure 4). In run2 our…"

Line 390: "model captures…" rather than "module captures…"?

Line 398: Why don't you include this energy in your calculations? Shouldn't' you? Would it be too difficult?

Line 416: Your ocean under modern day is incredibly fresh, and I am shocked that the ORA is that fresh. It can't be. Can you please recheck your calculations of ORA. It should be around 34.5 psu.

Line 432: Please include a citation or citations for your statement that the behaviour is expected.

Line 446: Please include a citation or citations for your statement that the behaviour is expected.

Line 446: Why is this linked to excess precipitation in SPEEDY? Seems the opposite of what I'd expect.

Line 462: Your correlations are not close to one. Please be more honest about this.

Figure 10: Can you provide a new column of differences? The differences in Antarctica are hard to see.

Line 493: Are these runs done with an IS model at equilibrium? If so, surely the response at 360 ppm of an IS model at equilibrium would be a good reason for the simulated flux of freshwater and ice sheet loss to be much greater than the observations? Please explain and make clearer here what was done.

Final paragraph before conclusions: I find it really interesting and actually a bit disappointing that you don't' showcase how the model could simulate a climate tipping point to a new regime. This hasn't been shown even though you talk about it in the introduction.

Line 535: "agree broadly with…"

---

## Author Comment (AC1)

**Response to Reviewer 1**

We sincerely thank the reviewer for the detailed report and for the constructive criticisms, which will be very useful for improving our manuscript. Before addressing each comment in detail, we would like to briefly respond to the main points and clarify our overall strategy.

In the current version, we focused exclusively on steady-state examples, in which the ice sheet evolves toward equilibrium with the corresponding climate and vegetation. This approach is particularly suitable for investigating deep-time climates, such as those, for example, of the Early Triassic, where paleogeographic reconstructions provided by PANALESIS or paleoMap do not include ice sheets or account for isostatic adjustments. Nevertheless, the climate conditions in the Early Triassic could be sufficiently cold to allow ice sheet formation, as demonstrated in previous numerical experiments conducted within our group (Ragon et al., 2024). Thus, it is relevant to have a tool that allows to estimate extent and volume of potential ice sheets in such cases. Despite the long response times of ice sheets to climate fluctuations we feel our steady-state approach is justified since for these deep-time climates, we do not have the (relatively) high temporal resolution that we have for more recent epochs.

However, before applying our coupled setup to deep-time climate configurations, it is essential to verify whether the model is capable of reproducing a plausible ice sheet that is in steady state with the preindustrial climate. We use the term "plausible" to acknowledge that the preindustrial ice sheet was not strictly in steady state with the climate as continental ice sheets have response times of thousands of years, and preindustrial climate fluctuations are over a shorter time scale than this, but still we expect the steady states to be at least qualitatively similar. Therefore, we focus on the preindustrial ice sheet starting from the bedrock topography (*run1*: preindustrial (PI) state).

Regarding *run2*, which represents how the ice sheet changes for a larger CO2 value until a steady state is reached, we agree with the reviewer that this experiment cannot be directly compared to observations, as the latter reflect transient, forced conditions. The purpose of *run2* was to verify that the modelled responses of the climate, ice sheet, and vegetation to a warming scenario are consistent with those obtained for the PI state (in terms of ice sheet and vegetation, in particular). As expected, the equilibrium state reached at 360 ppm CO2 is warmer than reanalysis data.

We therefore agree with the reviewer that including a transient simulation would be more appropriate for comparison with forced conditions. In the proposed transient experiment, the ice sheet is kept fixed at its PI configuration, while the responses of the climate and vegetation are examined under a gradual increase in atmospheric CO2 concentration from 280 ppm to 360 ppm. This transient simulation can be readily performed with our coupled setup and will be compared to available reanalysis data. Such an approach is commonly employed in CMIP-style CO2-forced experiments.

Such procedure can be applied to any arbitrary initial ice sheet configuration. We have shown in *run1* and *run2* that the ice-sheet model provides a realistic reconstruction of the ice sheet starting from the bedrock topography. This capability is particularly valuable for deep-time simulations, where the presence of ice sheets can be hypothesized but is not always constrained by proxy data (as in the case of Early Triassic climate oscillations). It is also relevant for experiments in which the external forcing is increased beyond certain thresholds, allowing us to explore the potential future behavior of ice sheets under extreme climate conditions and the associated evolution of biomes. These situations are complementary to those investigated using highly-resolved ice sheets, as in Smith et al. JAMES 2021.

The main advantages of our setup are the following:

- 1. The ice sheet model is global, which is particularly important for deep-time simulations where data are scarce and the potential location, extent, and thickness of ice sheets under cold scenarios are not known a priori.
- It includes the first-order, large-scale dynamics of ice sheets, allowing global-scale simulations to be performed without the need for high spatial resolution and with relatively low computational costs.
- 3. It employs the same cubed-sphere grid as the underlying climate model (MITgcm), thereby avoiding interpolation errors.
- 4. The ice sheet model is capable of representing the retreat and advance of ice sheets over shallow ocean regions. Vegetation grows on ice-free land and is replaced when the ice sheet expands.

Due to the simplified land module implemented in MITgcm, small-scale ice sheet dynamics cannot be fully represented. The more sophisticated STREAMICE module available in MITgcm is designed for regional applications at kilometer-scale resolution and has not been adapted to, or tested in, cubed-sphere coordinates. It also solves a higher-order stress balance, which would not make sense to use at our intended resolution, and steps explicitly in time. The effort to disentangle this higher-order stress balance, as well as implement the implicit time-stepping required to stabilize the shallow-ice balance would be significant. To overcome this limitation, we have directly developed an ice sheet module capable of running globally on the cubed-sphere grid rather than adopting another.

Finally, we fully agree with the reviewer that including a deep-time simulation is relevant to demonstrate the capability of our coupled setup to be applied to different paleogeographic configurations. We will therefore include the case of the Permian–Triassic cold state described in Ragon et al. (2024), in which we employed the ice sheet model to estimate the potential thickness and extent of the ice sheet that could develop in the Northern polar region.

In summary, the revised manuscript will include the following new simulations:

- 1. A transient simulation, starting from the PI state and forced by an increase in atmospheric CO2 concentration up to 360 ppm, to be compared with reanalysis data;
- 2. A Permian–Triassic simulation, corresponding to the cold-state configuration described in Ragon et al. (2024).

A reference MITgcm PI simulation (coupling atmosphere and ocean dynamics, with present-day ice sheet and fixed biomes) is presented in Brunetti & Vérard (2018). We will repeat this simulation using the same setup parameters adopted in the present study, in order to compare the resulting steady-state climate with our reconstructed ice sheet, vegetation, and climate fields in *run1*. The MITgcm is generally used in its ocean-only configuration within the ECCO program, while the SPEEDY module has been tested independently in atmosphere-only simulations (Molteni Clim. Dyn. 2003).

Based on the reviewer's comments, we acknowledge the need to improve the *Introduction* and to define more explicitly what we mean by "transient simulations" in our study.